# Polyglycerol-opioid conjugate produces analgesia devoid of side effects

Sara González-Rodríguez[1,2†‡], Mohiuddin A Quadir[3†§], Shilpi Gupta[3†¶], Karolina A Walker[3], Xuejiao Zhang[3], Viola Spahn[1], Dominika Labuz[1], Antonio Rodriguez-Gaztelumendi[1], Martin Schmelz[4], Jan Joseph[5], Maria K Parr[5], Halina Machelska[1,2], Rainer Haag[2,3], Christoph Stein[1,2*]

[1]Department of Anesthesiology and Critical Care Medicine, Charité Campus Benjamin Franklin, Freie Universität Berlin, Berlin, Germany; [2]Helmholtz Virtual Institute Multifunctional Biomaterials for Medicine, Teltow, Germany; [3]Institute of Chemistry and Biochemistry, Freie Universität Berlin, Berlin, Germany; [4]Department of Anesthesiology, Medical Faculty Mannheim, Universität Heidelberg, Heidelberg, Germany; [5]Institute of Pharmacy, Freie Universität Berlin, Berlin, Germany

*For correspondence: christoph. stein@charite.de

[†]These authors contributed equally to this work

Present address: [‡]Instituto de Biología Molecular y Celular, Universidad Miguel Hernández, Elche, Spain; [§]Department of Coatings and Polymeric Materials, North Dakota State University, Fargo, United States; [¶]Chapman University School of Pharmacy, Irvine, United States

Competing interests: The authors declare that no competing interests exist.

**Abstract** Novel painkillers are urgently needed. The activation of opioid receptors in peripheral inflamed tissue can reduce pain without central adverse effects such as sedation, apnoea, or addiction. Here, we use an unprecedented strategy and report the synthesis and analgesic efficacy of the standard opioid morphine covalently attached to hyperbranched polyglycerol (PG-M) by a cleavable linker. With its high-molecular weight and hydrophilicity, this conjugate is designed to selectively release morphine in injured tissue and to prevent blood-brain barrier permeation. In contrast to conventional morphine, intravenous PG-M exclusively activated peripheral opioid receptors to produce analgesia in inflamed rat paws without major side effects such as sedation or constipation. Concentrations of morphine in the brain, blood, paw tissue, and in vitro confirmed the selective release of morphine in the inflamed milieu. Thus, PG-M may serve as prototype of a peripherally restricted opioid formulation designed to forego central and intestinal side effects.

## Introduction

Opioid agonists such as morphine (*Figure 1*; 1) are the gold standard for the treatment of severe pain. However, their clinical effectiveness is limited by adverse side effects that mainly result from permeation of the blood-brain barrier (BBB) and activation of mu-opioid receptors in the central nervous system (CNS). These include sedation, apnoea, tolerance, and addiction and have recently lead to an epidemic of overdoses, death, and abuse (*Kolodny et al., 2015*; *Passik, 2014*). In addition, constipation due to activation of central and/or intestinal opioid receptors is a major drawback (*Schumacher et al., 2015*), and nonsteroidal analgesics are burdened by detrimental side effects such as gastrointestinal bleeding, ulcers, or cardiovascular complications (*Bhala et al., 2013*; *Arfè et al., 2016*; *Sondergaard et al., 2017*). Thus, novel painkillers are urgently needed.

Our approach is based on opioid receptors expressed and functional in peripheral sensory neurons (*Stein et al., 1990*; *Stein and Machelska, 2011*). Such receptors are upregulated in injured tissue and can be targeted selectively to avoid centrally mediated side effects. The local application of small, systemically inactive doses of opioid agonists has been demonstrated to potently reduce pain and inflammation in animal models and humans (*Stein and Machelska, 2011*; *Kalso et al., 2002*; *Vadivelu et al., 2011*; *Stein et al., 1991*; *Stein and Küchler, 2013*; *Graham et al., 2013*; *Zeng et al., 2013*), and pharmacological, genetic, and clinical studies have shown that a large proportion of the analgesic effects produced by systemically administered opioids can be mediated by

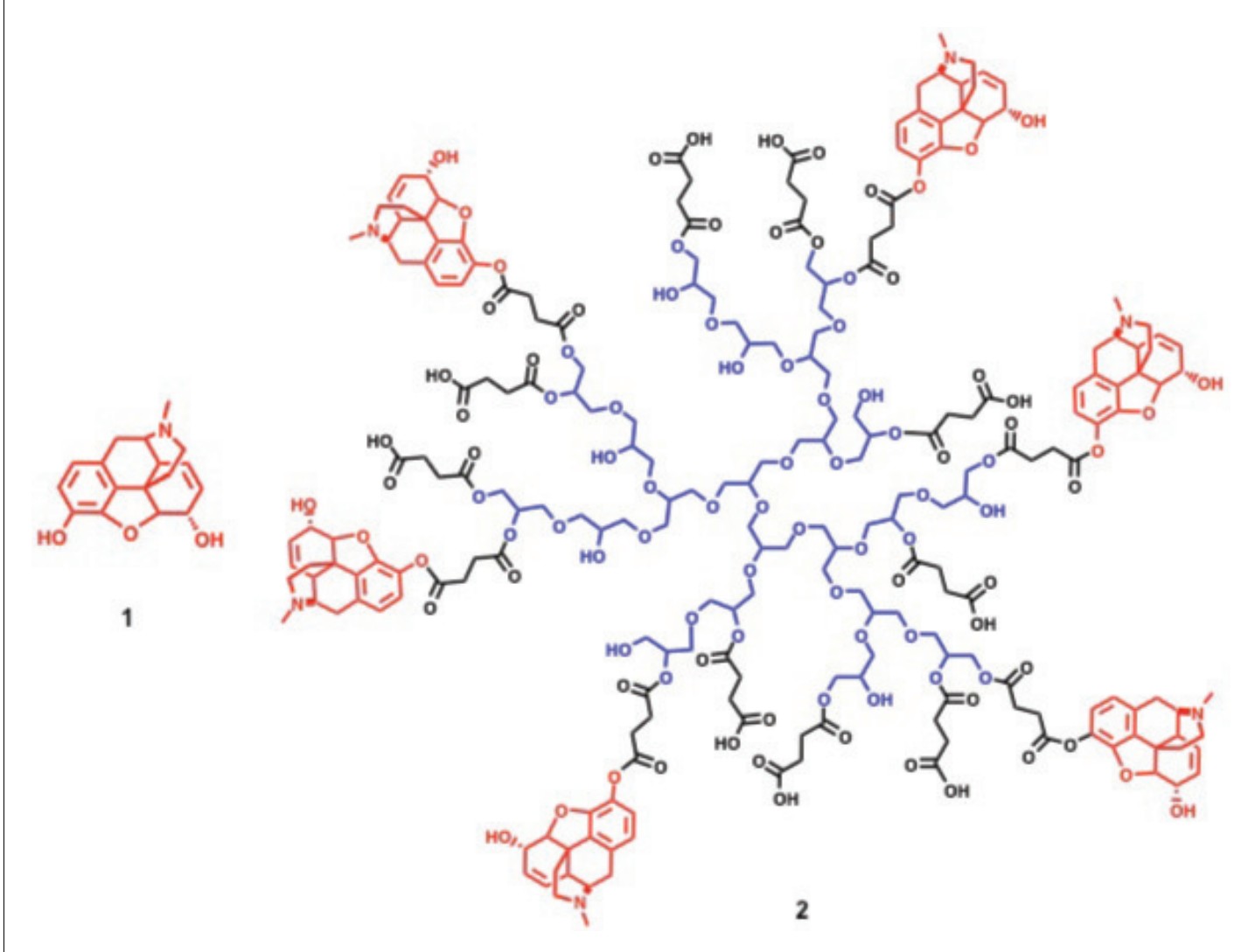

**Figure 1.** Structures of morphine (**1**) and PG-M (**2**).

peripheral opioid receptors (*Labuz et al., 2007*; *Gaveriaux-Ruff et al., 2011*; *Weibel et al., 2013*; *Jagla et al., 2014*; *Stein and Jagla, 2014*). However, peripherally restricted formulations of opioid drugs for systemic administration are not available so far.

Besides pharmacodynamic strategies to selectively activate opioid receptors in injured environments (*Spahn et al., 2017*), pharmacokinetic approaches have pursued polarized opioid ligands to decrease BBB permeability (*Stein and Machelska, 2011*; *Vadivelu et al., 2011*; *Stein and Küchler, 2013*; *Stein, 1993*) or carriers engineered for the targeted delivery of drugs, including antibody conjugates (*Hua and Cabot, 2013*; *Cheng et al., 2012*). The former were unsuccessful due to peripheral side effects (e.g. loperamide-induced constipation), or because polar residues can decrease ligand affinity to opioid receptors and do not achieve complete BBB impermeability at higher doses (*Stein and Machelska, 2011*; *Brown and Goldberg, 1985*). The latter are burdened by allergic reactions to proteins, instability in the circulation, low drug loading capacity, cross-reactivity with healthy or off-target tissues, impaired release of active drug, rapid elimination, or high molecular complexity (*Cheng et al., 2012*; *Baker and Carr, 2010*; *Koshkaryev et al., 2013*).

Here, we present an unprecedented strategy that does not aim at site-directed drug delivery via tissue-specific antigens or receptors, but harnesses the characteristics of the most prevalent form of clinical pain, that is, pain associated with localized or disseminated inflammation. Inflammation is

accompanied by aggregation of leukocyte esterase, proteases, low pH and other mediators, and is an essential component of multiple acute and chronic painful syndromes, including arthritis, endometriosis, cystitis, neuropathic pain, cancer, and surgery (*Kominsky et al., 2010*; *Holzer, 2009*; *Yu et al., 2015*; *Rindler-Ludwig et al., 1974*). Our approach relies on the high molecular size and hydrophilicity of a polyglycerol-morphine (PG-M) conjugate that hinders BBB permeation, on the enhanced permeability and retention (EPR) characteristic which permits diffusion of high-molecular-weight species through leaky blood vessels in inflamed tissue (*Khandare et al., 2012*; *Azzopardi et al., 2013*), and on the use of an ester bond as a cleavable linker (*Cheng et al., 2012*; *Koshkaryev et al., 2013*). We hypothesized that PG-M accumulates in inflamed tissue and releases morphine, which then produces analgesia *via* selective activation of peripheral but not central opioid receptors.

## Results

### Synthesis and characterization of polyglycerol-morphine (PG-M)

Based on our previous studies on PG-based targeting of inflammation and tumors (*Gröger et al., 2013*; *Calderón et al., 2011*), we constructed covalent PG-M conjugates with a size of 5 nm where morphine is immobilized on hyperbranched PG scaffolds by an ester linkage (*Figure 1*; 2). Hyperbranched PG is a novel class of hydrophilic dendritic macromolecules with multiple hydroxyl functional groups and a polyether backbone, characterized by tunable end group functionalities, defined topological 3D architecture, enhanced stability, and inertness to non-specific interactions with biological environments (*Calderón et al., 2010*; *Wilms et al., 2010*).

The PG-M conjugate (2) was obtained by a two-step protocol from morphine (1) and hyperbranched PG. The conjugate was purified by dialysis against phosphate buffered saline (PBS; pH 7.4). UV-visible spectra of dialyzed PG-M showed the absorption peak of morphine at 285 nm indicating successful immobilization of the drug on the PG scaffold (*Figure 2*). $^1$H-NMR revealed the characteristic appearance of proton resonance signals for morphine in the conjugated product (*Figure 3*). Physical encapsulation of morphine by PG was ruled out by thin-layer chromatography (TLC) using Dragendorff and Ninhydrin as selective reagents for identifying free morphine (*Figure 4*), and by a control reaction of PG with morphine-free base without coupling reagents. PG-M was dissolved in pH 7.4 and injected through a gel permeation chromatography (GPC) column with refractive index (RI) detection using water as a mobile phase. The conjugate eluted as a single pure species (*Figure 5*) with a polydispersity index (PDI) of 1.12.

Although elemental analysis is not a method of choice for quantification of drug-conjugated hyperbranched polymers (particularly of glycerol origin), mostly due to the hygroscopicity of the material, such compositional analysis was used to show the increment of nitrogen content (solely due to morphine) in PG-M. The N/C ratio increased from 0.018 ± 0.01% (PG-succinate) to 7.31 ± 0.21% (PG-M), that is, about 70-fold due to the immobilization of morphine-free base onto the PG backbone. One unit of PG-M contained 7.31% morphine-free base (*Table 1*).

### PG-M produces analgesia selectively in inflamed tissue

We induced unilaterally localized inflammation by intraplantar (i.pl.) injection of complete Freund's adjuvant (CFA) into one hindpaw of rats, and measured tissue pH as well as mechanical paw pressure thresholds (PPT) required to elicit hindlimb withdrawal (*Machelska et al., 1998*). This model is used widely and is of high predictive validity for acute and chronic pain syndromes (e.g. surgery, arthritis), likely because mechanical stimuli predominately determine human pain intensity (*Stein and Machelska, 2011*; *Kalso et al., 2002*; *Stein et al., 1991*; *Jagla et al., 2014*; *Stein, 1993*; *Le Bars et al., 2001*; *Whiteside et al., 2008*). We found significantly decreased pH (6.82 ± 0.02 vs. 7.20 ± 0.03; p<0.05, $t$ test, N = 9) and PPT (hyperalgesia) in the inflamed paw (i.e. lower baseline PPT in inflamed vs. noninflamed paws; see *Figure 6*, *Figure 7*), in line with numerous previous studies (reviewed in *Stein and Machelska (2011)*; *Stein (1993)*). We determined the effects of morphine or PG-M injected i.pl. into inflamed paws. The amount of morphine per mass of unit measure PG-M was quantified by UV-spectrophotometry and the dosages were calculated to contain the same absolute quantity of morphine (0–400 μg, calculated as the free base) per administration (*Table 1*). Dose-dependent (0–100 μg) PPT elevations (analgesia) were detected in inflamed but not in contralateral

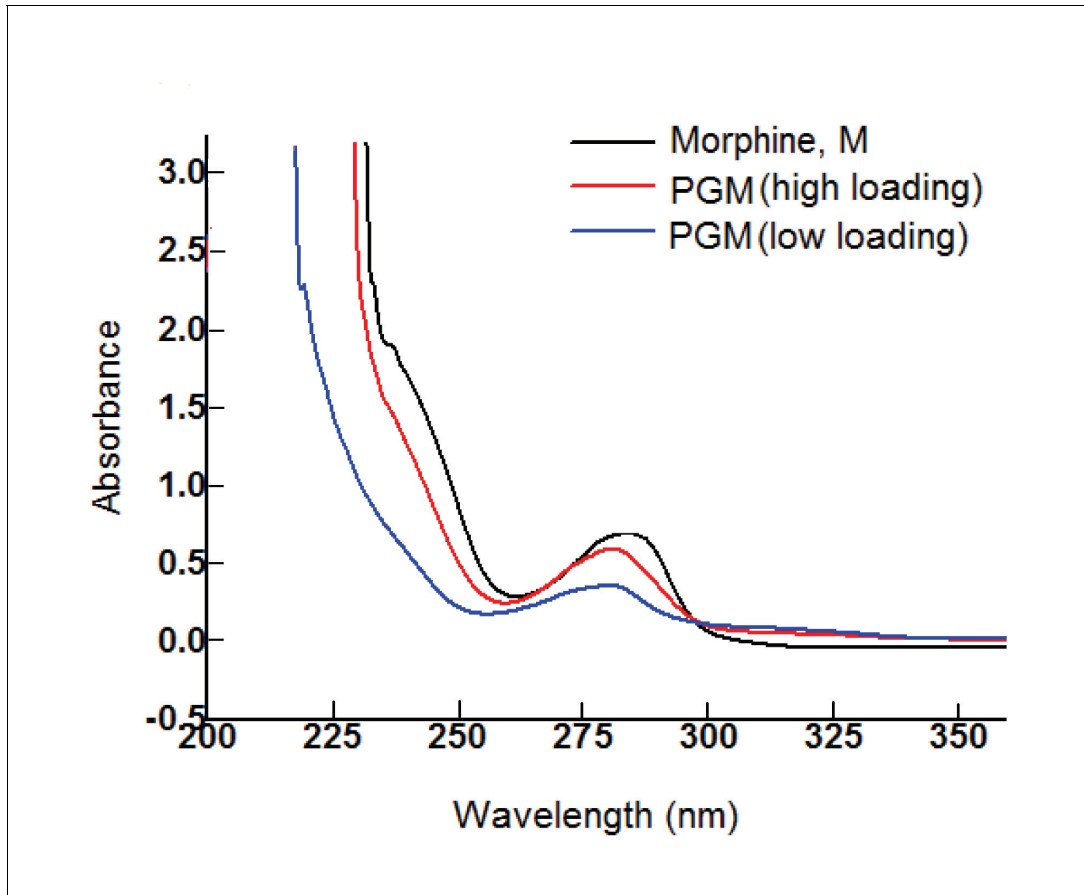

**Figure 2.** UV-visible spectrum of dialyzed PG-M showing the characteristic signal of morphine at 285 nm indicating the presence of the morphine molecule in its active form within the conjugate structure. Amounts of morphine in all chemical experiments were calculated based on UV-visible quantification using a calibration curve generated from free morphine. The UV spectrum for each sample was acquired using 30 scans per sample for maximized S/N ratio, and represents N = 3 experimental replicates.

noninflamed paws after injections of morphine (*Figure 6a,c*) or PG-M (*Figure 6b,d*). At a dose of 400 µg, i.pl. morphine evoked significant analgesia in both paws (*Figure 6a,c*) suggesting its systemic absorption and subsequent action in the CNS, whereas no contralateral effect was detected with the equivalent dose of PG-M (incorporating 400 µg of morphine-free base) (*Figure 6d*). Naloxone-methiodide (NLXM; 50 µg i.pl.), an opioid receptor antagonist unable to cross the BBB (*Brown and Goldberg, 1985*), completely blocked the maximum analgesic effects in inflamed paws induced by either i.pl. morphine or PG-M (equivalent to 100 µg morphine free base), demonstrating activation of peripheral opioid receptors by both agonists (*Figure 6e,f*). When the most effective dose (equivalent of 100 µg morphine free base) of either compound was injected unilaterally into noninflamed paws, PG-M did not alter thresholds but morphine significantly elevated PPT in treated paws (*Table 2*). The latter effect was blocked by NLXM (50 µg i.pl.). Together, these data suggest that locally administered PG-M is effective exclusively at peripheral opioid receptors in injured tissue, while morphine can activate opioid receptors in central, as well as in injured and healthy peripheral compartments.

Intravenous (i.v.) morphine produced dose-dependent analgesic effects in both inflamed and non-inflamed paws (*Figure 7a,c*). At 12 mg/kg morphine, the cut-off PPT (250 g) was reached in both paws, accompanied by overt sedation and decreased respiratory frequency. These findings are consistent with previous studies using similar doses of morphine (*Nakamura et al., 2011*; *Bajic et al., 2015*). In contrast, i.v. PG-M (up to the equivalent of 6 mg/kg morphine-free base; *Table 1*) produced dose-dependent analgesia selectively in inflamed paws, but did not elicit stronger effects at

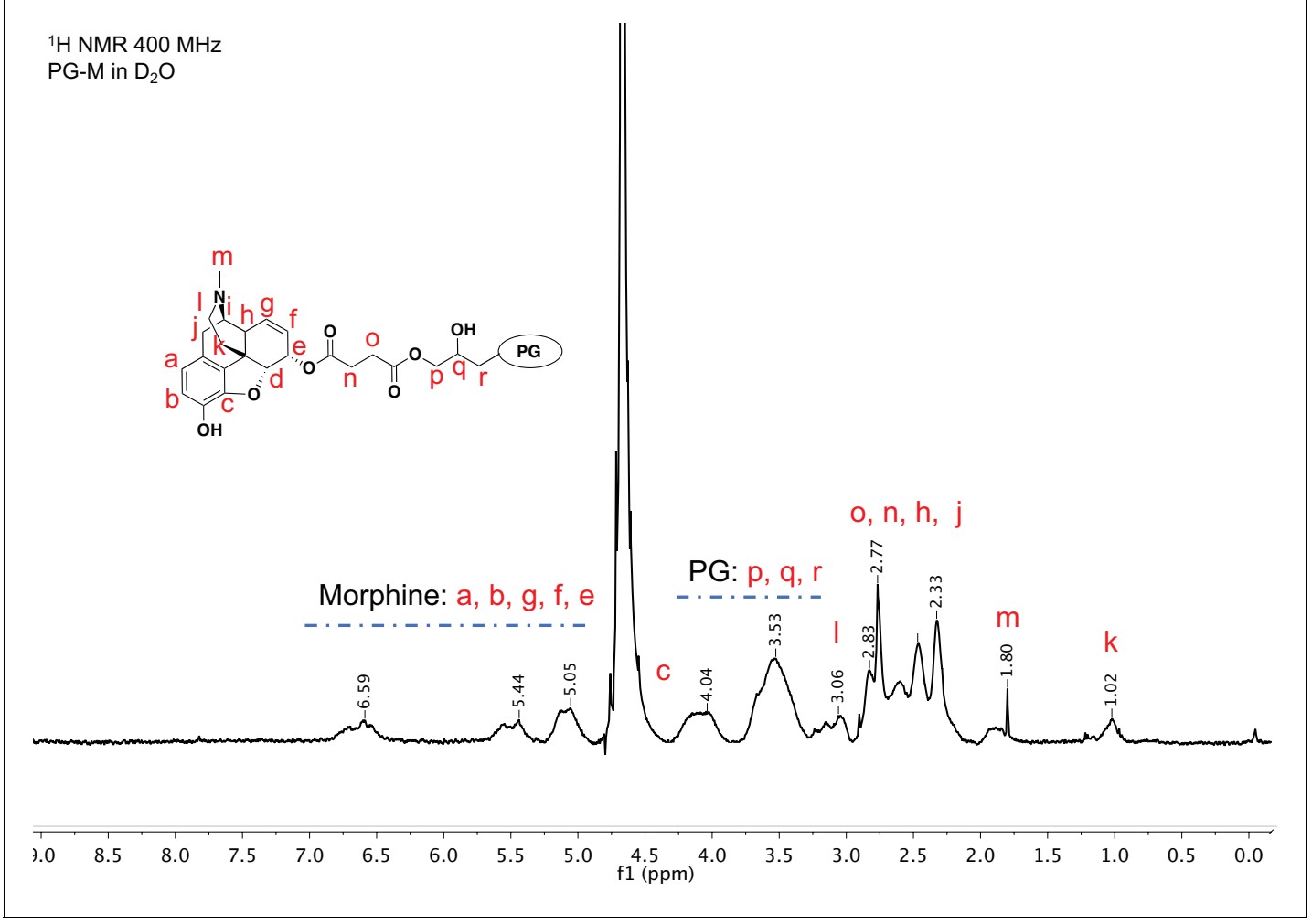

**Figure 3.** [1]H nuclear magnetic resonance (NMR) spectroscopy of purified PG-M showing resonance signals from morphine along with those from the PG scaffold, indicating successful conjugation of the small molecule morphine on hyperbranched PG. PG-M was purified by dialysis and size exclusion chromatography. NMR of the lyophilized product shows signals from aromatic protons of morphine from 5.1 to 6.6 ppm. Further protons from the conjugate molecule are also assigned to the spectrum. There was no evidence for the presence of free morphine salt or any other small molecular impurities in the sample. The spectral acquisition is a result of N = 32 scans to optimize S/N ratio, and the representative spectrum is obtained from N = 3 experiments.

12 mg/kg morphine-free base equivalent, or any changes in noninflamed paws (*Figure 7b,d*). The bilateral i.pl. injection of NLXM (50 μg/paw) did not influence the analgesic effect of 6 mg/kg i.v. morphine in noninflamed paws, and reversed the effect in inflamed paws only partially (by about half), suggesting the involvement of both central and peripheral opioid receptors (*Figure 7e*). In contrast, analgesia induced by i.v. PG-M (equivalent to 6 mg/kg morphine free base) in inflamed paws was completely blocked by NLXM (50 μg/paw i.pl.; *Figure 7f*), suggesting that only peripheral opioid receptors were activated.

## PG-M produces no central sedative effects or constipation

Four days after induction of unilateral paw inflammation, we administered i.v. morphine (12 mg/kg), PG-M (12 mg/kg morphine-free base equivalent) or NaCl and recorded the animals' locomotor activity in the open field by an automated camera. The distance travelled was determined in 5 min intervals for 60 min. Rats receiving morphine were markedly sedated, similar to previous reports (*Nakamura et al., 2011*; *Trujillo et al., 2011*). Their activity was significantly lower than that of animals treated with PG-M or NaCl, while no differences between the latter two groups were

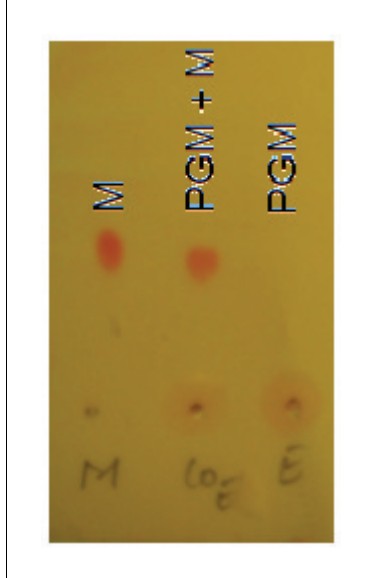

**Figure 4.** Thin-layer chromatography (TLC) of PG-M showing retention of morphine (M) at baseline on normal-phase silica plate (lane 3), and no evidence of non-covalently entrapped morphine within the conjugate, as indicated by positive Dragendorf and Ninhydrin test. Representative chromatograph obtained from N = 3 experiments.

detectable at any time (*Figure 8a*). To assess constipation, separate groups received i.v. morphine, PG-M (each at 12 mg/kg morphine free base equivalent) or NaCl, and fecal boli were collected for 60 min. Morphine completely suppressed bowel movements (similar to other studies [*Nakamura et al., 2011*]), whereas number or total weight of fecal boli were not significantly different between rats treated with PG-M or NaCl (*Figure 8b*). Together, our data on analgesic, locomotor, and intestinal effects indicate that i.v. PG-M exclusively acts at peripheral opioid receptors in injured tissue. In contrast, i.v. morphine ubiquitously activates peripheral and central opioid receptors. A rough estimate based on i.v. morphine-induced PPT elevations and their reduction by NLXM (*Figure 7e*) suggests that up to one half of morphine's analgesic effect is mediated peripherally, similar to studies in human patients (*Jagla et al., 2014*).

## In vivo concentrations of free morphine in paw tissue, circulating blood, and brain

To directly examine whether morphine is released from PG-M conjugates in subcutaneous paw tissue, we collected microdialysates in vivo and compared these to blood and brain samples after administration of the most effective agonist doses established in behavioral experiments. Morphine concentrations were determined by enzyme-linked immunosorbent assay (ELISA) and pH was obtained in venous blood (7.37 ± 0.01; N = 6 rats with paw inflammation; 7.36 ± 0.01; N = 6 controls). After unilateral i.pl. injections of morphine or PG-M (both equivalent to 100 µg morphine free base) into inflamed paws, we detected up to 1200 ng/ml or 576 ng/ml of morphine in paw dialysates, respectively (*Figure 9a,b*). In contralateral noninflamed paws, morphine (about 19 ng/ml at 1 hr) was only detectable after i.pl. administration of morphine but not of PG-M. Following 6 mg/kg i.v. morphine, we measured mean peak morphine concentrations of up to 1700 ng/ml in inflamed paws (*Figure 9a*), 247 ng/ml in noninflamed paws, 1500 ng/ml in blood (*Figure 9c*), and 15 ng/ml in brain (*Figure 9d*). After i.v. PG-M (equivalent to 6 mg/kg morphine-free base), we obtained up to 115 ng/ml morphine in inflamed paws (*Figure 9b*) but no relevant concentrations of free morphine in noninflamed paws (*Figure 9b*), blood or brain (*Figure 9e*) at any time.

## In vitro opioid receptor binding, cytotoxicity and morphine release

In radioligand-binding studies using human embryonic kidney (HEK) 293 cells stably expressing rat mu-opioid receptors (*Spahn et al., 2013*), PG-M was 10,000 times less effective than morphine to displace 4 nM [$^3$H]-([D-Ala2, N-MePhe4, Gly-ol]-enkephalin) (DAMGO), a standard mu-opioid receptor ligand. This was demonstrated by a marked rightward shift of the concentration-response curve and a vast difference in IC$_{50}$ values (*Figure 10*), indicating that PG-M does not bind to mu-opioid receptors. Effects on cell viability were not significantly different between morphine and PG-M, as measured by the MTT cell proliferation assay and real-time cell analysis (RTCA) (*Figure 11*). To study in vitro cleavage in more detail, we synthesized a conjugate containing 48% morphine-free base per unit of PG-M, as determined by $^1$H NMR spectroscopy. High-performance liquid chromatography hyphenated to tandem mass spectrometry (LC-MS/MS) revealed that separate exposure to either human leukocyte esterase (pH 7.4) or to pH 5.5 released 36% or 10%, respectively, of the initially conjugated morphine in a time-dependent manner (*Figure 12*).

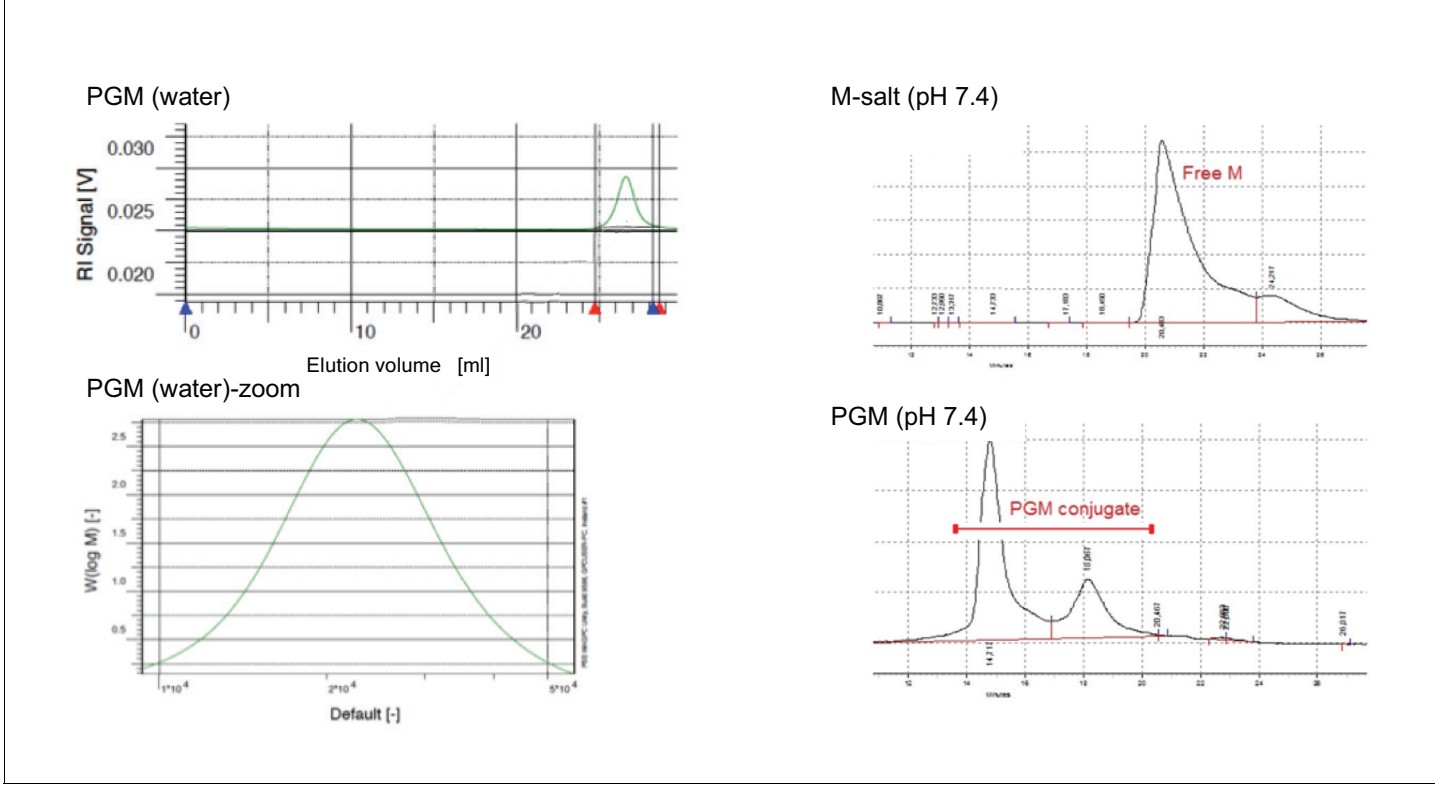

**Figure 5.** Gel permeation chromatogram (GPC) of morphine and PG-M. Left: GPC of PG-M using water as eluent shows the monodispersed signal of the pure conjugate with a PDI of 1.12 at a flow rate of 1 ml/min (N = 3 experimental replicates). The average molecular weight of PG-M was 21.4 kDa. This permits low-molecular-weight fractions, originating most likely from the conjugation of morphine (M) to low-molecular-weight PG remaining from the initial anionic polymerization process for the synthesis of the hyperbranched scaffold. Due to the statistical nature of molecular weight distribution of hyperbranched polymers, it was not possible to obtain a monomodal MALDI signal of PG or PG-M. The bottom excerpt shows zoom into PG-M peak. Right: GPC of M-salt (top) and PG-M (bottom), incubated in buffer without enzymes. Measurements were performed using BioSil SEC250 columns (Biorad) with a mobile phase flow rate of 1 ml/min under isocratic elution. An injection volume of 50 µl (of either M-salt or PG-M) was used in a mobile phase composed of 10% acetonitrile, 90% 10 mM sodium phosphate buffer, 0.15 M NaCl, pH 7.0, and detection at 285 nm. Free unmodified M (dimeric morphine sulfate) starts eluting between 20 and 22 min (top), while PG-M begins eluting at 14 min and does not yield free M (bottom). The second peak following the major PG-M signal derives from lower molecular weight conjugates of PG-M (bottom).

## Discussion

Chronic diseases with an inflammatory component are the greatest health threat (*Tabas and Glass, 2013*) and inflammation is an essential component of multiple painful syndromes such as arthritis, wounds, endometriosis, cystitis, cancer or metastatic bone pain related to osteoclasts (reviewed in *Stein and Machelska (2011)*; *Holzer (2009)*). Since inflammation and pain can be effectively inhibited by activation of peripheral opioid receptors (*Stein and Machelska, 2011*; *Vadivelu et al., 2011*; *Stein and Küchler, 2013*; *Zeng et al., 2013*; *Gaveriaux-Ruff et al., 2011*; *Weibel et al., 2013*; *Jagla et al., 2014*; *Spahn et al., 2017*), we designed conjugates to target bioactive compounds toward injured tissue (*Fleige et al., 2012*; *Wang et al., 2007*). Dendritic PG-based molecules are characterized by high hydrophilicity, low systemic toxicity, prolonged circulation half-life, reduced BBB permeability, and high on-target accumulation (*Baker and Carr, 2010*; *Khandare et al., 2012*; *Calderón et al., 2010*). The solubility profile of PG can be controlled by the functionalization gradient among and within the surface hydroxyl groups (*Haag et al., 2000*). By conjugation with hydrophobic free morphine base at a low functionalization level (<10%), we expected PG-M to retain its inherent hydrophilicity. We converted PG hydroxyl groups to succinate functionalities, which were esterified with morphine through carbodiimide-mediated esterification. This compound released morphine when exposed to either leukocyte esterase or acidic pH in vitro. Because polymer conjugates can selectively aggregate in tumor tissue based on the EPR effect

**Table 1.** Conversion of PG-M amounts into morphine-free base equivalents.

**Intraplantar administration**

| PG-M amount (µg) | Morphine-free base equivalent (µg) | % morphine-free base per unit PG-M |
|---|---|---|
| 5471.95 | 400 | 7.31 |
| 1367.98 | 100 | 7.31 |
| 683.99 | 50 | 7.31 |
| 341.99 | 25 | 7.31 |

**Intravenous administration**

| PG-M amount (mg/kg) | Morphine-free base equivalent (mg/kg) | % morphine-free base per unit PG-M |
|---|---|---|
| 164.15 | 12 | 7.31 |
| 82.07 | 6 | 7.31 |
| 54.72 | 4 | 7.31 |
| 27.35 | 2 | 7.31 |
| 13.68 | 1 | 7.31 |
| 6.84 | 0.5 | 7.31 |

(*Khandare et al., 2012*; *Azzopardi et al., 2013*), we assumed that the hyperpermeable vasculature in inflamed tissue would permit a targeted accumulation of PG-M and subsequent release of free morphine in vivo.

Using a standard model of high predictive validity for clinical analgesia (*Stein and Machelska, 2011*; *Kalso et al., 2002*; *Stein et al., 1991*; *Stein, 1993*; *Machelska et al., 1998*; *Le Bars et al., 2001*; *Whiteside et al., 2008*), we found that both conventional morphine and PG-M produced stronger analgesic effects in inflamed than in noninflamed tissue. This is consistent with numerous experimental and clinical studies and has been attributed to upregulation, increased functionality and accessibility of peripheral opioid receptors (*Stein and Machelska, 2011*; *Kalso et al., 2002*; *Spahn et al., 2017*). After local injection of conventional morphine or PG-M into inflamed paws, peak morphine concentrations in microdialysates coincided with peak analgesic effects within 5–20 min. These concentrations were about twice as high after conventional morphine than after PG-M, even though we administered the same absolute amounts of morphine. This may be due to protracted release of morphine from the conjugate. In both cases, the analgesic effect in inflamed paws was completely blocked by NLXM, indicating peripheral opioid receptor activation. The effects of both compounds were of similar magnitude, suggesting that maximum efficacy is determined by the available number of peripheral opioid receptors rather than by the amount of agonist, in line with our previous observations after recruitment of opioid-producing inflammatory cells (*Brack et al., 2004*). Upon local injection of morphine into inflamed paws, small amounts of morphine and a minor analgesic effect appeared in contralateral noninflamed paws (*Figure 6c*), suggesting vascular absorption of morphine and subsequent stimulation of central opioid receptors.

I.v. morphine produced dose-dependent analgesia in both hindlimbs. NLXM partially reduced this response in inflamed but not in noninflamed paws, indicating that the effect in inflamed paws was mediated by both central and peripheral, while that in noninflamed paws was mediated by central opioid receptors only. Concentrations of free morphine measured in brain, blood, and paw tissue corroborated these results. Considering the low microdialysis perfusion rate (see below), the morphine levels in subcutaneous paw tissue within the first hour correlated well with analgesic actions, consistent with the notion that peripheral opioid receptors mediated these effects. Enhanced plasma extravasation most likely accounted for the higher morphine concentrations in inflamed than in noninflamed paws.

At dosages containing equivalent amounts of morphine-free base, i.v. PG-M produced equally strong analgesia as morphine, but restricted to the inflamed paw. This effect was completely reversed by NLXM, indicating exclusive peripheral opioid receptor activation. Consistently, we did not detect free morphine in noninflamed paw tissue, blood, or brain. This suggests that the

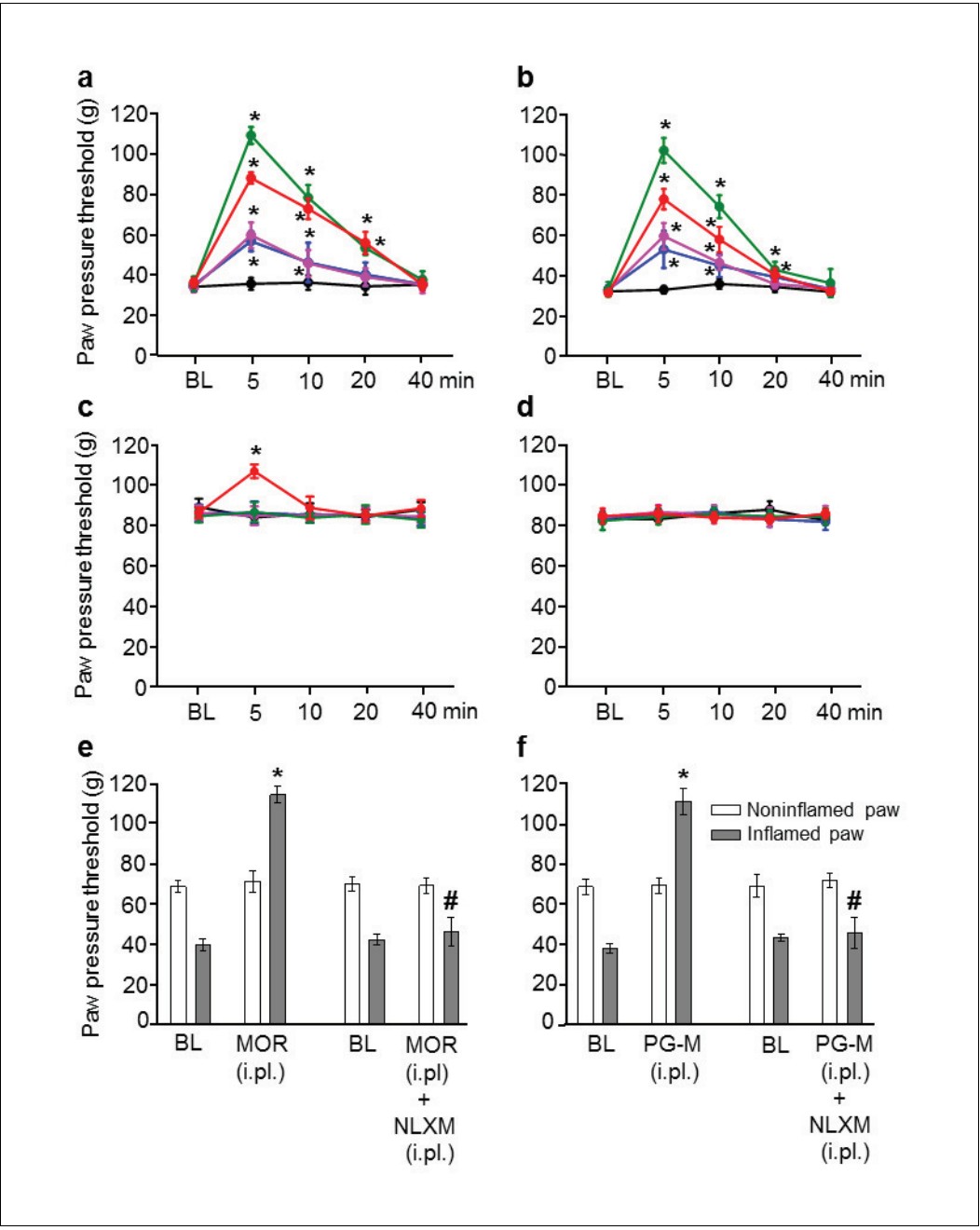

**Figure 6.** Analgesic effects after intraplantar (i.pl.) injection of morphine or PG-M in rats with unilateral hindpaw inflammation. Effects on mechanical pain thresholds (PPT) in inflamed (**a, b**) and noninflamed (**c, d**) hindpaws following unilateral i.pl. injections of morphine (MOR; **a, c, e**) or PG-M (**b, d, f**) into the inflamed paw. Dosages of MOR and PG-M represent the absolute amounts of morphine-free base (see **Table 1**). Unilateral i.pl. injection of NLXM (50 µg/paw) completely reversed peak PPT elevations occurring at 5 min after i.pl. morphine (100 µg; **e**) or i.pl. PG-M (equivalent to 100 µg morphine; **f**). BL: baseline PPT before injections. (**a, b, c**) *$p<0.05$, two-way RM-ANOVA and Bonferroni test (compared to controls). (**e, f**) *$p<0.05$, paired t-test (BL vs. agonist); #$p<0.05$, unpaired t-test (agonist vs. agonist + NLXM). N = 8 rats per group; means ± SEM. PPT increases in a and b (0–100 µg) were dose-dependent ($p<0.05$, linear regression ANOVA).

The following source data is available for figure 6:

**Source data 1.** Raw data for **Figure 6**.

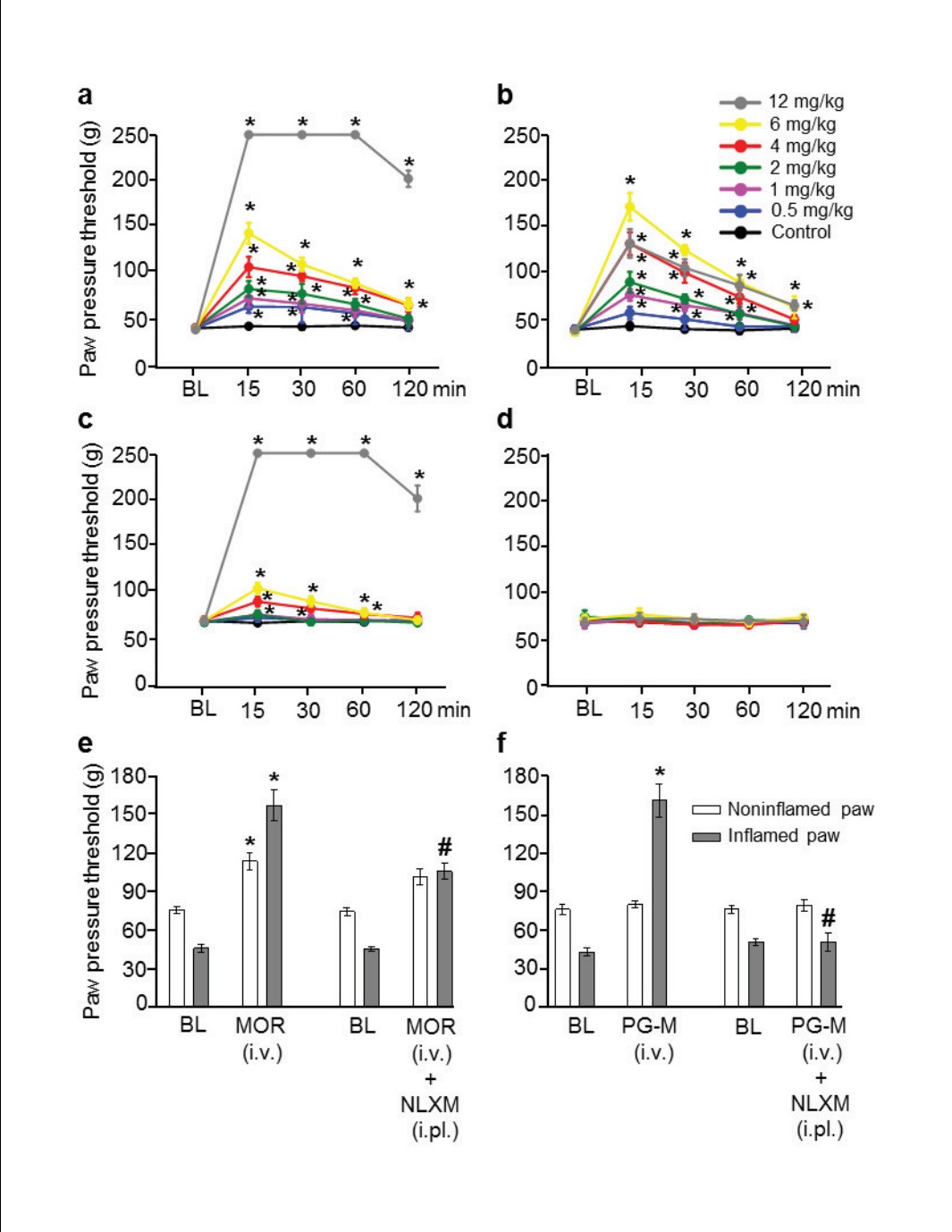

**Figure 7.** Analgesic effects after intravenous (i.v.) morphine or PG-M in rats with unilateral hindpaw inflammation. Effects on mechanical pain thresholds (PPT) in inflamed (**a, b**) and noninflamed (**c, d**) hindpaws following i.v. injections of morphine (MOR; **a, c, e**) or PG-M (**b, d, f**). Dosages of MOR and PG-M represent the absolute amounts of morphine free base (see *Table 1*). Bilateral intraplantar (i.pl.) injection of NLXM (50 µg/paw) partially reduced peak PPT elevations occurring at 15 min after MOR (6 mg/kg; **e**) and completely abolished effects after PG-M (equivalent to 6 mg/kg morphine; **f**). (**a, b, c**) *p<0.05, two-way RM-ANOVA and Bonferroni test (compared to controls). (**e, f**) *p<0.05, paired t-test (BL vs. agonist without/with NLXM); #p<0.05, unpaired t-test (agonist vs. agonist with NLXM). N = 8 rats per group; means ± SEM. PPT increases in a, c (0–12 mg/kg) and b (0–6 mg/kg) were dose-dependent (p<0.05, linear regression ANOVA).

The following source data is available for figure 7:

*Figure 7 continued on next page*

*Figure 7 continued*

**Source data 1.** Raw data for *Figure 7*.

conjugates are not cleaved in the absence of inflammation and are unable to cross the BBB. The maximum analgesic effect preceded the peak concentrations of morphine in paw microdialysates. However, such a delay is expected since morphine is cleaved at the injured site where it can readily bind local opioid receptors, and only the spill-over morphine may diffuse to the dialysis membrane. The low perfusion rate of 1 µl/min will cause an additional delay of at least 10 min for the dialysate to reach the outlet. Thus, given a paw volume of about 1–2 ml (*Machelska et al., 1998*), the first peak at the outflow should not occur before 35–45 min after PG-M extravasates into subcutaneous paw tissue. Our data indicate that after i.v. administration, PG-M is retained in the circulation, slowly accumulates at the target site and produces prolonged analgesia, likely based on the EPR effect (*Khandare et al., 2012*; *Azzopardi et al., 2013*). After i.pl. administration, the analgesic effect was of shorter duration, consistent with the notion that PG-M does not need to extravasate and is cleaved soon after injection.

No free morphine was detectable in blood after i.v. PG-M, suggesting that this conjugate is stable in absence of leukocyte esterase at physiologic pH. Our direct measurements revealed pH-values of about 6.8 in inflamed paws in vivo. Free morphine was measurable in vitro when PG-M was exposed separately either to low pH or to leukocyte esterase. The in vitro release occurred at a relatively slow rate with a clear enzymatically driven profile. In addition to acidosis and leukocyte esterase, the combination of multiple inflammatory mediators and other lytic enzymes such as proteases may promote a much faster cleavage of PG-M in vivo. These factors appear to contribute to the selective release of morphine from PG-M in injured but not in normal environment, as encountered in brain, blood, or intestinal wall. Indeed, while conventional morphine produced sedation both in the open field and in PPT experiments (rats receiving high doses frequently reached cut-off values), our microdialysis, locomotor, analgesia and binding assays clearly indicated that PG-M did not reach the CNS and did not bind to opioid receptors. Consistent with the notion that PG-M should not extravasate or be cleaved in the noninjured intestinal wall, we did not detect any signs of constipation, in contrast to previously developed peripheral opioid agonists (e.g. loperamide). The abuse potential and organ toxicity of PG-M need to be investigated in further detail, although free hyperbranched PG has already been declared safe for in vivo application (*Kainthan et al., 2006*).

**Table 2.** Mechanical pain thresholds at 5 min after unilateral intraplantar injections of morphine or PG-M into the noninflamed paw. Dosages of morphine and PG-M represent the same absolute amounts of morphine-free base (100 µg; see *Table 1*). Unilateral injection of NLXM (50 µg/paw i.pl.) into the noninflamed paw completely reversed morphine-induced PPT elevation. PG-M did not significantly change PPT in the noninflamed paw. *p<0.05, unpaired t-test (compared to 0.9% NaCl). #p<0.05, unpaired t-test (compared to morphine +0.9% NaCl); N = 8 rats per group; means ± SEM.

| Treatment | Paw pressure thresholds (PPT) | |
|---|---|---|
| | Noninflamed paw | Inflamed paw |
| Control (0.9% NaCl) | 67.3 ± 1.9 | 40.2 ± 1.1 |
| Morphine | 94.4 ± 1.7* | 43.3 ± 1.4 |
| Control (morphine + 0.9% NaCl) | 86.8 ± 3.0* | 41.4 ± 2.0 |
| Morphine + NLXM | 71.5 ± 3.1# | 42.9 ± 1.0 |
| Control (0.9% NaCl) | 68.3 ± 2.1 | 43.0 ± 1.1 |
| PG-M | 69.9 ± 0.6 | 42.5 ± 1.0 |

**Source data 1.** Raw data for *Table 2*.

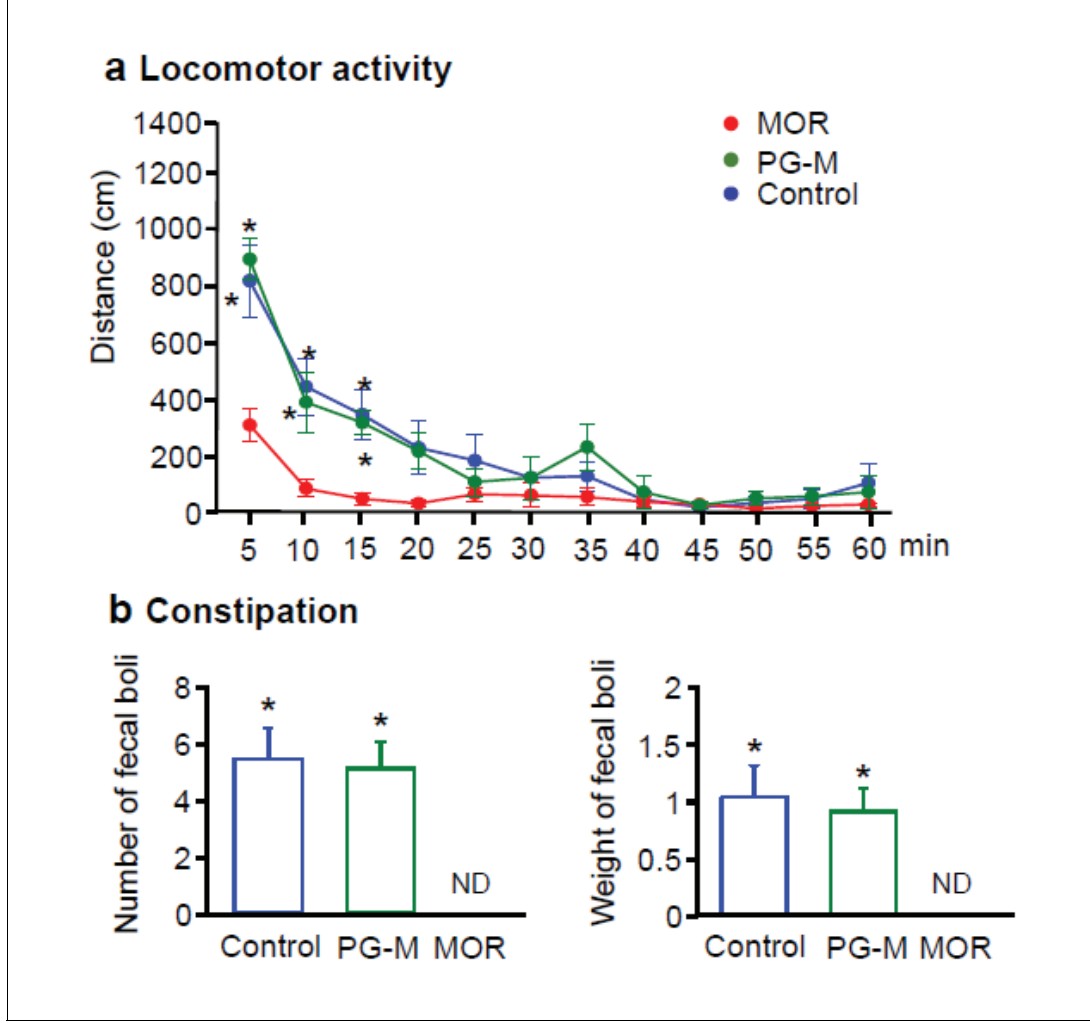

**Figure 8.** Effects of intravenous (i.v.) morphine or PG-M on locomotor activity and defecation in rats with unilateral hindpaw inflammation. (**a**) Locomotor activity after i.v. injections of morphine (MOR), PG-M or 0.9% NaCl. *$p<0.05$, two-way RM-ANOVA and Bonferroni test (compared to MOR); N = 8 rats per group; means ± SEM. (**b**) Number (left) and total weight (right) of fecal boli produced after i.v. injections of MOR, PG-M or 0.9% NaCl. Dosages of MOR and PG-M represent the same absolute amounts of morphine-free base (12 mg/kg; see *Table 1*). ND: none detected; *$p<0.05$, one-way ANOVA and Bonferroni test (compared to MOR); N = 8 rats per group; means ± SEM.

The following source data is available for figure 8:

**Source data 1.** Raw data for *Figure 8*.

---

In summary, our concept offers a novel perspective for the design of safer opioid analgesics by use of nanocarriers and suggests additional areas for exploration, for example the development of conjugates with other opioids or with antihistamines to preclude central sedative effects.

## Materials and methods

### Materials

All chemicals and solvents were of analytical grade and purchased from Sigma-Aldrich GmbH and Fisher Scientific GmbH. We synthesized hyperbranched PG according to published procedures (*Sunder et al., 1999*, *Sunder et al., 2000*; *Haag et al., 2002*) ($M_n$ = 10,000 g/mol; PDI = 1.7). All PG samples were concentrated and dried under vacuum (50°C, $1 \times 10^{-2}$ mbar) until loss of weight

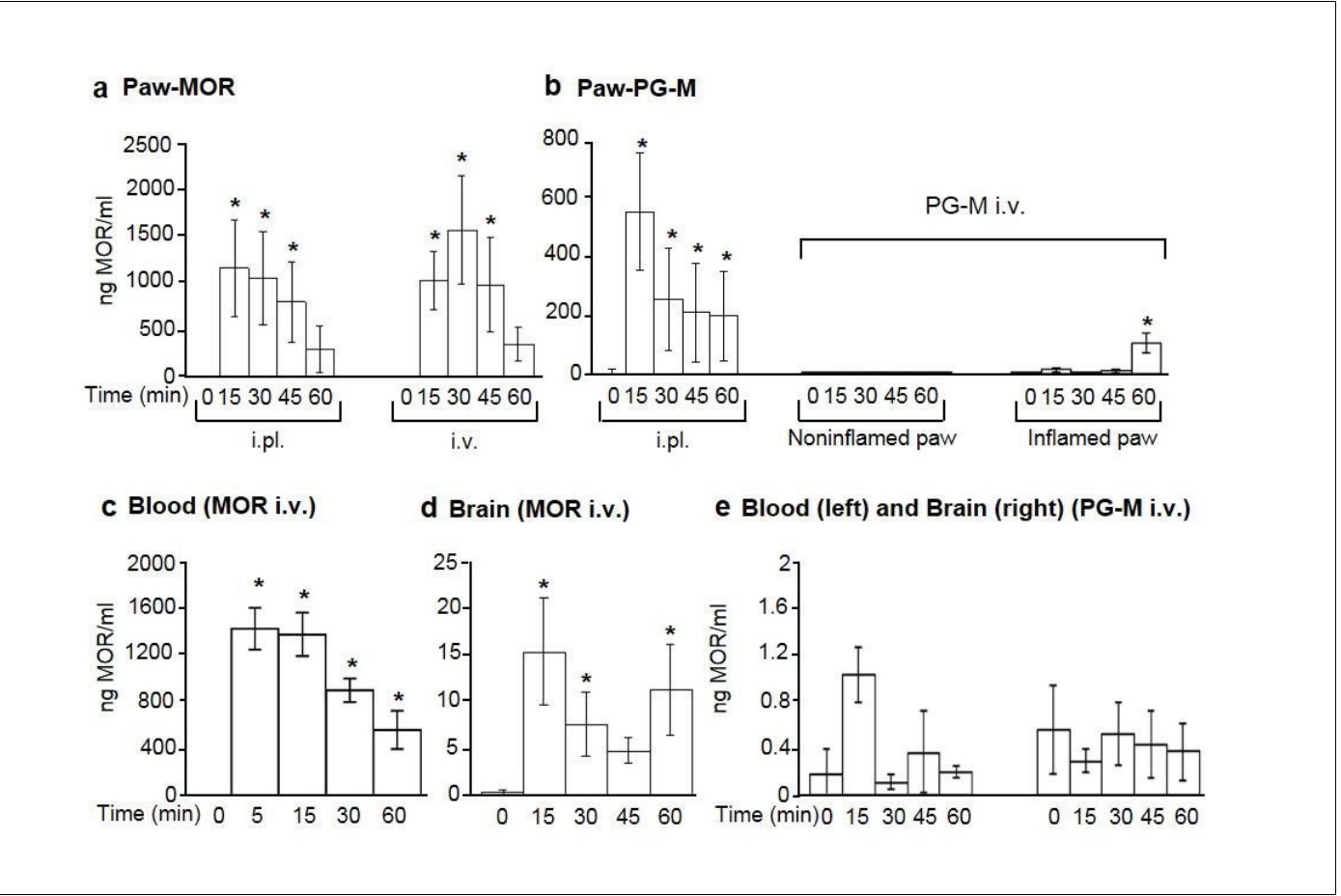

**Figure 9.** Concentrations of free morphine in paw microdialysates, blood, and brain. Concentrations in paw microdialysates after intraplantar (i.pl.; 100 μg morphine free base equivalent; N = 7) or intravenous (i.v.; 6 mg/kg morphine free base equivalent; N = 7) administration (at time 0) of morphine (MOR) (a) or PG-M (b). Blood and brain concentrations after i.v. injection of morphine (c, N = 8; d, N = 7) or PG-M (e; N = 6 for blood; N = 7 for brain) (6 mg/kg morphine-free base equivalent in both cases). Each sample was divided for duplicate measurements by ELISA. (a, b, c) *p<0.05, one-way RM-ANOVA and Bonferroni tests (compared to corresponding values before injections, that is, at 0 min). (d) *p<0.05, Kruskal-Wallis one-way ANOVA on ranks and Dunn test (compared to values before injection); means ± SEM.

The following source data is available for figure 9:

**Source data 1.** Raw data for *Figure 9*.

was lower than 0.025 g per 1.0 g of the dried sample in 5 hr drying periods. This process was accepted as standard procedure. Analytical TLCs were performed on precoated Merck silica-gel 60F254 plates with dichloromethane: methanol:ammonia (18:2:0.1 v/v/v) as the mobile phase, visualized either by UV or by using Ninhydrin and Dragendorff reagents for staining. Milli-Q water (resistance ~18 MΩ.cm, pH = 5.6 ± 0.2; Merck Millipore) was used in all experiments and for preparation of all samples. Dialysis was performed using benzoylated dialysis tubing (pore size 2,000 Da; Sigma-Aldrich), changing the solvent three times over a period of 24 hr. For determination of morphine release in vitro, centrifuge filter devices (pore size 3,000 Da; Merck Millipore) and native human leukocyte esterase (suspension in 154 mM NaCl; Creative Enzymes) were used. Analytic reference material (morphine-HCl) was of *Pharmacopoeia Europaea* quality (Fagron, Barsbüttel, Germany). Mobile phase solvents and additives were of liquid chromatography–mass spectrometry (LC-MS) grade (VWR, Darmstadt, Germany). Fresh ultrapure water was obtained from a LaboStar 2-DI/UV system (SG Wasseraufbereitung und Regeneration, Barsbüttel, Germany) equipped with LC-Pak

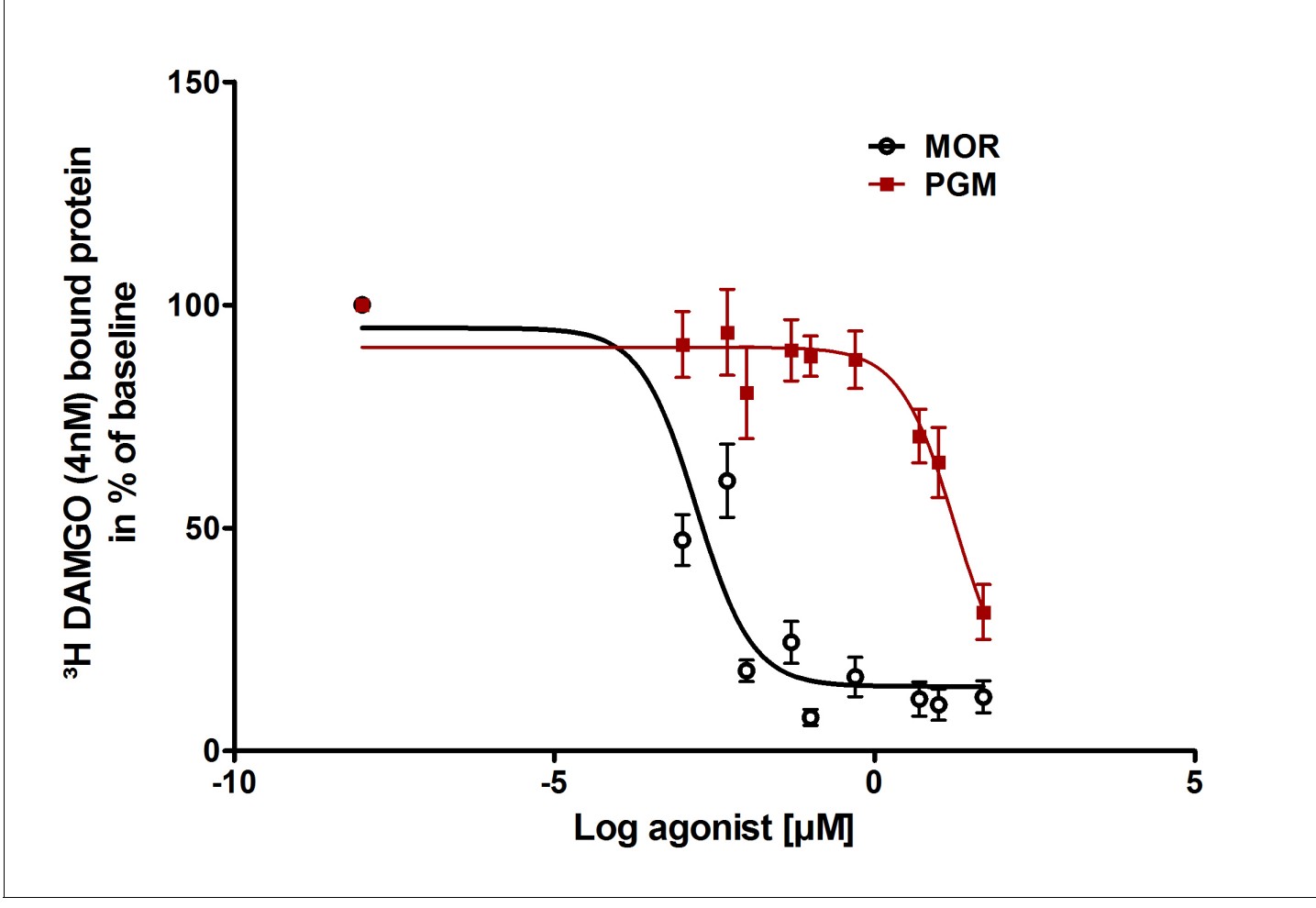

**Figure 10.** Binding of PG-M and morphine to mu-opioid receptors. Different concentrations of morphine (MOR; circles) or PG-M (squares) were applied to compete with 4 nM [3 hr]-DAMGO binding to mu-opioid receptors stably expressed in HEK 293 cells. The affinity of PG-M was negligible compared to morphine ($IC_{50}$: 18.7 ± 1.82 μM vs. 0.002 ± 1.37 μM; p<0.001, unpaired t-test); means ± SEM of six independent experiments, each performed in duplicates.
The following source data is available for figure 10:

**Source data 1.** Raw data for *Figure 10*.

Polisher and a 0.22 μm membrane point-of-use cartridge (Millipak). All synthetic and physico-chemical experiments were performed in triplicates by different experimenters.

## Instrumentation

The $^1$H and $^{13}$C NMR spectra were recorded on a Bruker DRX 400, and a Bruker AMX 500 MHz spectrometer (Bruker GmbH). The spectra were calibrated using the solvent peaks. The chemical shift values were on δ scale and the coupling constant values (*J*) in Hz. Molecular weight and molecular weight distribution ($M_w/M_n$) of PG were determined using GPC equipped with Agilent 1100 pump, refractive index detector, and PLgel and Suprema columns. The polymers were eluted with MilliQ water at a flow rate of 1.0 ml/min. Molecular weights were calibrated with pullulan standards. UV-absorption spectra were recorded between 220 and 800 nm using a Scinco S-3150 UV-vis spectrophotometer (Scinco Co. Ltd.) (range: 187–1193 nm; resolution: 1024 points). All measurements were carried out in a thermostated UV-cell (1 cm). The LC-MS/MS instrument was comprised of an Agilent 1290 II UHPLC with Infinity 1290 II autosampler (kept at 4°C) coupled to an Agilent 6495 triple quadrupole LC-MS/MS system with Agilent Jet Stream electrospray ionization (ESI) source and ion funnel

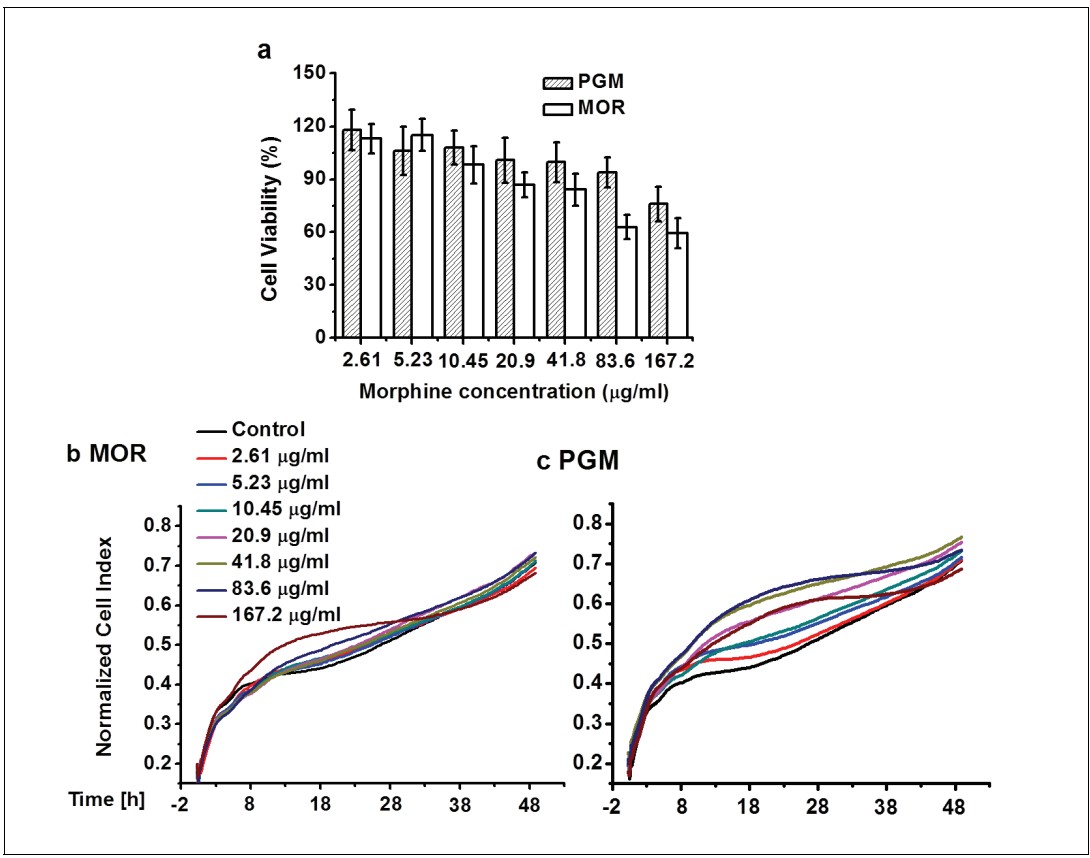

**Figure 11.** Cytotoxicity. Viability of NIH 3T3 cells determined by MTT assay (**a**). Differences between morphine and PG-M were not significant (p>0.05, Mann-Whitney U Test). RTCA cytotoxicity profiles of NIH 3T3 cells treated with morphine (**b**) and PG-M (**c**). Dosages were calculated to contain the same concentrations of morphine-free base (see captions) in both cases. Neither morphine nor PG-M decreased CI values.

(Agilent Technologies, Waldbronn, Germany). The MS was operated in positive ionization mode at a capillary voltage of 3000 V. A drying gas flow of 15 l/min at 180°C, sheath gas flow of 12 l/min at 385°C and a nebulizer pressure of 40 psi were used. MS/MS detection was performed in multiple reaction monitoring using the ion transitions 286.2→201.1 (quantifier, collision energy CE = 23V), 286.2→173.1 (CE = 28V), 286.2→165.0 (CE = 35V), and 286.2→153.0 (CE = 45V), the latter all as qualifiers. Separation was performed on an Agilent Zorbax C18 column (2.1 mm x 50 mm, 1.8 µm particle size, Agilent Technologies) at 30°C with a mobile phase of water containing 5 mM ammonium acetate (adjusted to pH 7.3 with aqueous NH3, eluent A) and methanol (eluent B) running a gradient (starting at 15% B, linear increase to 50% B in 1.4 min, followed by increase to 95% B until 2.5 min, 1.5 min hold, and final re-equilibration at 15% B for 1.5 min) at a flow rate of 400 µl/min and ambient temperature. Aliquots of 1 µl were injected onto the system.

## Isolation of free morphine base

Morphine sulfate pentahydrate (0.775 g, 1.02 mmol) was dissolved in minimum amount of Millipore water. The pH of the solution was adjusted to 9.2 with $NH_4OH$, where the free morphine base started precipitating as white flakes. The precipitate was centrifuged at 4000 rpm for 1 hr and dried over calcium chloride at room temperature to obtain free morphine base (0.145 g, 50%) (**Figure 1**; 1).

## Synthesis of PG succinate

PG (10 kDa, 3.0 g, 40.5 mmol OH groups) was dissolved in dry N, N´-dimethylformamide (DMF) (15 ml). Upon complete dissolution of PG, succinic anhydride (8.1 g, 81.0 mmol) was added to the reaction mixture. Subsequently triethylamine (5.6 ml, 40.5 mmol, 1.0 eq.) was added and the reaction

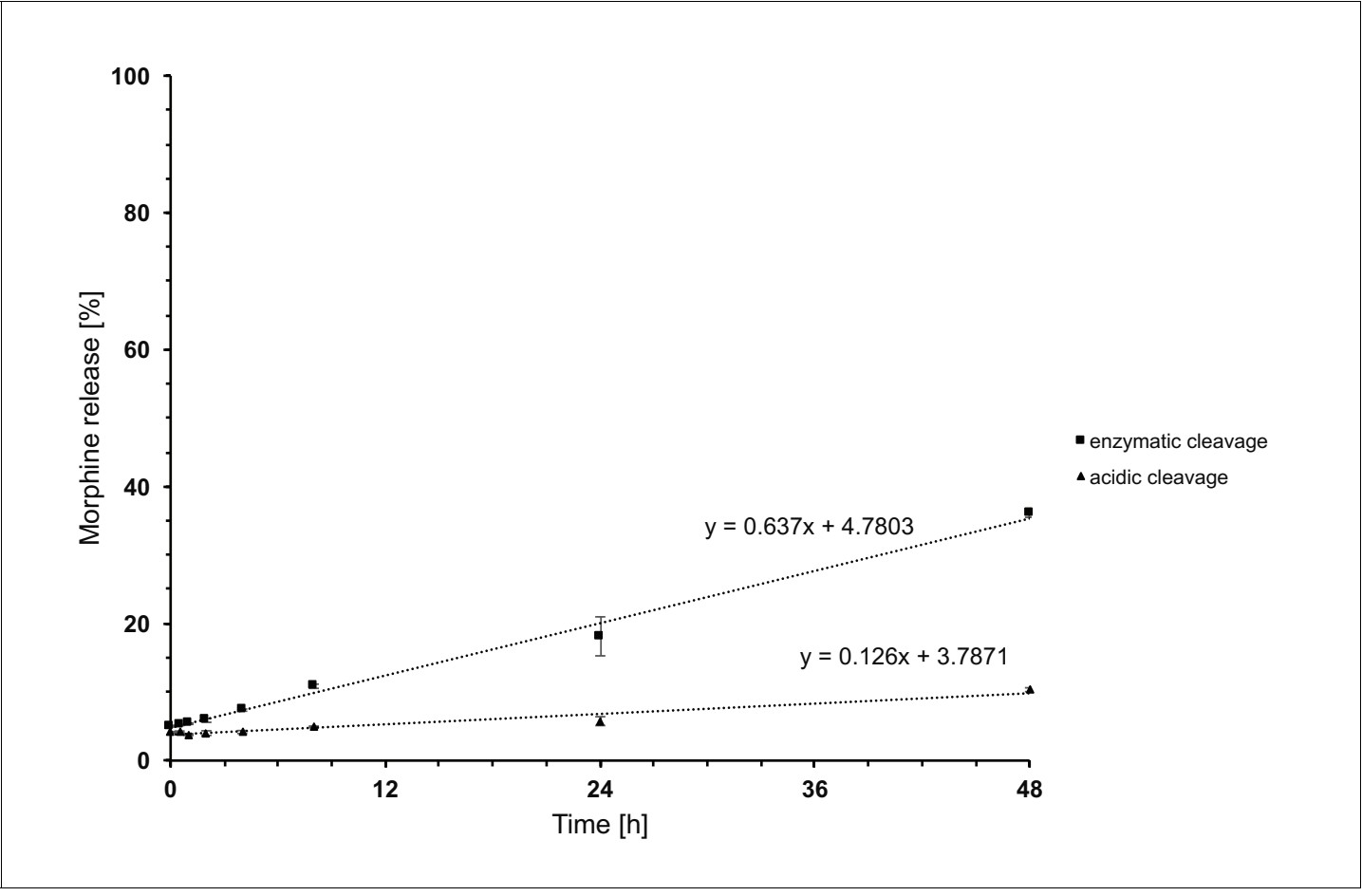

**Figure 12.** Time-dependent in vitro release of morphine from PG-M. The y-axis indicates percentage of free morphine relative to amount of morphine conjugated in PG-M, examined at 37°C in acidic solution (pH 5.5; triangles) or in the presence of human leukocyte esterase (pH 7.4; squares).

mixture was heated at 50°C for 48 hr. At the end of the reaction period, DMF was removed by cryo-distillation. An extensive dialysis of the crude product in water yielded PG succinate (5.92 g, 84%); conversion: quant. IR (KBr): 3672, 2876, 2751, 1787, 1707, 1632, 1575, 1213 cm-1; $^1$H NMR (500 MHz, CD3OD, δ): 5.33–5.12 (functionalized secondary PG-groups), 4.52–4.11 (functionalized primary PG-groups), 3.92–3.43 (PG-backbone), 2.81–2.52 (bs, 4 hr, -CH2-CH2-), 0.91 (PG-starter); $^{13}$C NMR: (CD3OD, δ): 171.3 (-COOH), 80.2–69.3 (PG), 28.5–26.6 (-CH2-CH2).

## Synthesis of PG-M

To a solution of PG succinate (0.5 g, 2.8 mmol) in dry DMF (25 mL), free morphine base (0.1g, 0.35 mmol) was added and stirred for 10 min. Consecutively, EDCI (0.07 g, 0.70 mmol) and DMAP (catalytic) were added with a mixing period of at least 5 min between the addition of two components. The reaction was allowed to run overnight. We followed the conjugate formation by TLC developed with Ninhydrin and Dragendorf stain. The course of the reaction was monitored by the gradual disappearance of the spot for free morphine base as well as retention of the colored conjugate at TLC baseline (*Figure 4*). After the specified reaction period, DMF was removed and the residue was dialyzed in phosphate buffer of pH 7.4 for no more than 8 hr, followed by size-exclusion chromatography performed on a Sephadex G 25 column with pH 7.4 buffer as eluent. The solution was freeze-dried to yield the desired product PG-M (*Figure 1*; 2) as white solid (0.628 g, morphine equivalent: 20.46 mg, morphine loading: 7.3%). IR (KBr) 2824, 1784, 1712, 1625, 1568, 1435, 1324 cm-1; $^1$H NMR (500 MHz, D$_2$O, δ): 6.81 (bs, Ar-H), 6.71 (bs, Ar-H), 5.47 (bs), 5.07 (bs), 4.40–4.00 (functionalized secondary PG-groups), 3.90–3.20 (functionalized primary PG-groups), 3.17–2.30 (-CH$_2$-C*H$_2$*-

COO, C$H_2$-CH$_2$-COO, -C$H_2$-CH-N-CH$_3$), 1.95–1.80 (-N-C$H_3$), 1.04–0.99 (-C$H_2$-CH$_2$-N-CH$_3$) (the detailed proton spectral assignment performed by comparing the proton signals with the resonance shift of the solvent peak at 4.79 ppm is shown in *Figure 3*), $\lambda_{max}$285 nm for conjugated morphine; elemental analysis of the conjugate yielded N/C (%)=7.33. GPC of the final conjugate was conducted in a Suprema column system using water as eluent at a flow rate of 1 ml/min with refractive index (RI) detection of $M_n$ = 21.404 g/mol, $M_w$ = 24.057 g/mol, PDI = 1.12 (*Figure 5*). For in vitro cleavage studies, PG-M with higher morphine content (48%) was synthesized analogously.

## Animal model of inflammatory pain

After approval by the state ethics committee (Landesamt für Gesundheit und Soziales, Berlin, Germany; No. G0115/09), experiments were performed in male Wistar rats (180–240 g; 5–7 weeks old; Janvier Laboratories). Animals were kept on a 12 hr light-dark schedule with food and water ad libitum. Room temperature was 22 ± 0.5°C and humidity 60–65%. Following i.pl. injection of 150 µl CFA (Calbiochem) into the right hindpaw under brief isoflurane (Abbott) anesthesia, rats developed an inflammation confined to the inoculated paw. All experiments were conducted at 4 days after inoculation of CFA. All further injections were performed under brief isoflurane anesthesia.

## In vivo pH measurement

A pH-sensitive glass microelectrode (model IC-401 combination pH electrode, Warner Instruments) was calibrated using reference solutions of pH 4.0, 7.0 and 9.2. Rats were anesthetized with isoflurane. A 20-gauge needle was used to pierce the plantar skin of both hindpaws. Thereafter, the microelectrode mounted in the lumen of a 20-gauge needle was advanced 3–6 mm deep into the subcutaneous tissue. Stable readings were obtained 2–3 min after insertion. In separate groups of rats with or without paw inflammation, pH values were obtained in venous blood from the right ventricle upon thoracotomy.

## Drugs and administration

The following drugs were used for in vivo behavioral experiments: morphine sulphate (7,8-didehydro-4,5a-epoxy-17-methylmorphinan-3,6a-diolsulfate) (Sigma-Aldrich), NLXM ((5a,17R)−4,5-Epoxy-3,14-dihydroxy-17-methyl-6-oxo-17-(2-propenyl)-morphinanium iodide; N-methylnaloxonium iodide) (Sigma-Aldrich). For measurements of morphine concentrations ex vivo, morphine-HCl (Merck) was used because morphine sulphate produced erratic results in ELISA. Morphine doses were calculated as the free base and drugs were dissolved in 0.9% NaCl. The amount of morphine per mass of unit measure PG-M was quantified by UV-spectrophotometry and the dosages were calculated to contain the same absolute quantity of morphine per in vivo administration (*Table 1*). Naloxone methiodide (NLXM) was administered 10 min before measurements of pain thresholds, that is, 5 min before i.pl. or 5 min after i.v. injections of morphine or PG-M. Control animals received vehicle (0.9% NaCl). Injection volumes were 100 µl (i.pl.) and 600 µl (i.v.). The experimenter was blinded to the drugs and doses applied. An assistant prepared drug solutions in separate coded syringes and then gave them to the experimenter in randomized order. The code was broken after completion of the experiment.

## Measurement of nociceptive thresholds

Paw pressure thresholds (PPT) were assessed using an algesiometer (Ugo Basile) by an experimenter blinded to the treatments (*Machelska et al., 1998*). Rats with unilateral hindpaw inflammation were handled for 4 days to accommodate them to testing. On the day of testing, rats were held under paper wadding, and incremental pressure was applied via a wedge-shaped, blunt piston onto the dorsal surface of the hindpaw by an automated gauge. The pressure required to elicit paw withdrawal (PPT) was recorded. A cut-off was set at 250 g to avoid tissue damage. The average of three trials, separated by 10 s intervals, was calculated. The same measurement was performed on the contralateral paw in alternated sequence to preclude order effects.

## Measurement of sedative effects and constipation

Locomotor activity in an open field was assessed using plastic cages with dark walls (44×44 × 40 cm, without top) (Ugo Basile) and the Any-Maze Video Tracking System (Stoelting Co.) connected to a computer for automated data analysis. The distance in cm travelled by each animal during 5 min

intervals was determined. After habituation on the day before testing, animals received i.v. injections of saline, PG-M or morphine, were individually placed into the cages and activity was measured for 60 min. In separate groups, fecal boli were collected for 60 min, counted and weighed according to published procedures (*Devilliers et al., 2013*).

## In vivo microdialysis

This was set up according to published procedures on in vivo microdialysis in the skin of humans and rats (*Sauerstein et al., 2000*; *Steinhoff et al., 2003*) and was modified to the plantar area of the hindpaw (*Antunes bras et al., 2001*). Rats were anesthetized with 100 mg/kg intraperitoneal pento-barbital (Sigma-Aldrich). To keep body temperature stable, we used a thermopad perfused with warm (38°C) distilled water by a pump (Gaymar industries). Oral temperature was recorded every 15 min with a digital thermometer (Hartmann) and was kept stable at 37°C throughout the experiment. Three single hollow plasmapheresis fibers (Asahi Medical Co.; 0.4 mm diameter, cut-off 3000 kDa) were inserted into the plantar subcutaneous tissue of the paw via 25-gauge hypodermic needles. The fibers were perfused with Ringer's solution via Tygon tubes (Novodirekt) attached by 26-gauge needles (Neoject) to 1 ml syringes (Dispomed) which were mounted on a microdialysis pump (Pump 22; Harvard Apparatus). After plantar passage, the fibers were inserted into glass capillaries (150 μl; 1 mm inner diameter; Servoprax R). The pump was set at a constant flow rate of 1 μl/min and 15 min later collection of dialysates into polyethylene cups was initiated. Another 15 min later (at time 0) morphine or PG-M were injected i.pl. or i.v.. The dialysates were collected over 15 min intervals as follows: 15 min before drug injections until time 0 (- 15 min – 0 min), 0 min – 15 min, 15 min – 30 min, 30 min – 45 min, and 45 min – 60 min. Dialysates were frozen at −80°C until determination of morphine content by ELISA.

## ELISA for morphine

Samples were processed according to the manufacturer's instructions (Abnova KA0935). To stay within the range of the standard curve, some samples (microdialysates after i.pl. injections, blood plasma after thawing) were diluted 1:100 with PBS. Optical density was measured by Spectra Max (Molecular Devices). Data were analyzed by the Softmax program. The manufacturer excluded cross-reactivities with morphine metabolites and related molecules.

## Sample preparation

Concentrations of free morphine in paw dialysates, blood, and brain were measured at different intervals after i.pl. or i.v. injection of morphine or PG-M, according to the occurrence of maximum analgesic effects in behavioral experiments. Dialysates were thawed and added directly into the ELISA plate wells. Blood samples were obtained by intracardiac puncture under deep isoflurane anesthesia, directly transferred onto ice and immediately afterwards spun down at 10°C and 3000 rpm for 7 min. Directly after centrifugation, the supernatant (plasma) was collected and stored at −80°C. To collect brain samples, rats were perfused under deep isoflurane anesthesia with saline, brain was removed and stored at −80°C. Once thawed, brain tissue was weighed and homogenized in 6 μl buffer per mg of tissue with Ultra-Turrax-T8 (IKA). The buffer consisted of 0.1 M Tris, 0.15 M NaCl, 0.1% CHAPS (Sigma-Aldrich) and a protease inhibitor (1 tablet/50 ml buffer, Roche Diagnostics). Homogenates were centrifuged at 15,000 g for 15 min at 4°C.

## In vitro release of morphine

Acidic cleavage was examined using 1.2 ml of PG-M solution (0.875 mg/ml) in sodium acetate buffer (1.5 M; pH 5.5). The solution was divided into 24 reaction tubes containing 50 μl each, and the tubes were placed in a heating block for incubation at 37°C and constant shaking at 500 rpm. At defined time points, the incubation was stopped by shock freezing in liquid $N_2$. Thereafter, all samples were thawed simultaneously at room temperature, 50 μl PBS solution (10 x PBS; pH 7.4) per sample was added, followed by the addition of 900 μl PBS solution (1 x PBS; pH 7.4), yielding a (theoretical) PG-M concentration of 0.0438 mg/ml. The samples were transferred to filter devices (pore size 3000 Da) and centrifuged for 20 min at 4000 U/min and 4°C. The filtrates were isolated for LC-MS measurements. Enzymatic cleavage was examined using 1.2 ml PG-M solution (0.875 mg/ml) in PBS buffer (pH 7.4) mixed with human leukocyte esterase (8.688 ml). The solution was divided into 24 reaction

tubes containing 50.4 µl each. Incubation and termination of the reaction was performed as described above. After thawing at room temperature, 50.4 µl PBS solution (10 x PBS; pH 7.4) was added, followed by 899 µl PBS solution (1x PBS; pH 7.4), yielding a (theoretical) PG-M concentration of 0.0434 mg/ml. Centrifugation was then performed as described above. In control experiments, PG-M solution (1 mg/ml) was incubated in PBS buffer (pH 7.4) without leukocyte esterase. All experiments were performed in triplicates.

## Cell culture and membrane preparation

An established standard cell line for the expression of exogenous proteins, HEK 293 (obtained from Deutsche Sammlung von Mikroorganismen und Zellkulturen; DSMZ, Braunschweig, Germany), stably expressing rat mu-opioid receptors was used. According to DSMZ, this cell line is negative for mycoplasms in DAPI, microbiological culture, mRNA hybridization and PCR assays, and multiplex PCR of minisatellite markers revealed a unique DNA profile confirmed as human using isoenzyme analysis of aspartate amino transferase and lactate dehydrogenase with isoelectric focusing gel electrophoresis. The cells were maintained in Dulbecco's Modified Eagle Medium (Sigma-Aldrich) supplemented with 10% fetal bovine blood plasma, 1% penicillin/streptomycin and 0.1 mg/ml geneticin (Biochrome) at 37°C and 5% $CO_2$ in a cell incubator. The cells were passaged 1:3 - 1:10 every second to third day depending on the confluence. For binding experiments, cells were cultured in flasks with a growth area of 175 cm$^2$. Cells were washed with ice-cold Trizma (50 mM, pH 7.4) (Sigma-Aldrich), scratched with a cell scraper, homogenized, and centrifuged twice at 42,000 g for 20 min at 4°C. Protein concentration was determined using the Bradford method.

## Radioligand binding

Displacement of the radiolabeled standard mu-opioid receptor agonist [3H]-DAMGO (4 nM) by different concentrations of morphine or PG-M (50, 10, 5, 1, 0.5, 0.1, 0.05, 0.01, 0.005 µM) was used to determine the half maximal concentration ($IC_{50}$) necessary to displace [3H]-DAMGO from the receptor. A protein amount of 80–100 µg was incubated with [3H]-DAMGO (50 Ci/mmol) and test compounds were dissolved in 50 mM Trizma at pH 7.4 for 90 min at room temperature. Nonspecific binding was determined by the addition of 10 µM naloxone hydrochloride, a standard opioid receptor antagonist (Sigma-Aldrich).

## Cytotoxicity

An established standard cell line for assessment of cytotoxicity, NIH 3T3 cells (DSMZ no.: ACC 59), were seeded into 96-well plates at 5000 cells per well. According to DSMZ, this cell line is negative for mycoplasms in DAPI, microbiological culture, mRNA hybridization and PCR assays, and it was confirmed as mus musculus by cytochrome oxidase I DNA barcoding. After 24 hr incubation, the culture medium was removed and replaced with 100 µl PG-M or morphine sulphate in DMEM at different concentrations. After 48 hr incubation, 10 µl of the tetrazolium dye 3-(4,5-dimethylthiazol-2-yl)−2,5-diphenyltetrazolium bromide (MTT solution; 5 mg/ml PBS) was added to each well. Cells were further incubated for 2 hr, culture medium was removed, and 100 µl DMSO were added to dissolve the purple formazan crystals. Optical density was measured at 490 nm by a microplate reader. Untreated cells were used as a control. The experiment was repeated three times and mean values are reported.

## Real-time cell analysis (RTCA)

This assay is a real-time technique to dynamically monitor cell viability. Microelectrode-coupled microtiter plates were used to culture cells, and the cell number, cell morphology, and degree of cell adhesion were reflected by the electrode impedance. Untreated cells attach to the bottom of the well and cause an increase of the cell index (CI) value due to the increased impedance. When treated with toxic compounds, the cells die and detach from the bottom, displaying a decrease of the CI value. 50 µl of culture medium was added to each well of the E-plate 96 (Roche) for background measurements. NIH 3T3 cells were then added at a concentration of 5000 cells per well for 24 hr incubation, followed by adding PG-M and morphine sulphate diluted with DMEM to contain morphine-free base concentrations of 2.6 to 168 µg/ml. Untreated cells were used as controls. The E-plate was incubated and monitored on the RTCA SP system (Roche) for 48 hr at intervals of 15 min

for 48 hr. The normalized CI was calculated by CI $_{original}$/CI $_{normalize\ time}$ using the RTCA software version 1.2.1. Four replicates were measured for each sample and the means are reported.

## Statistics

Effect sizes were not pre-specified and no *a priori* power analysis was run. Animal and sample numbers were chosen based on similar previous studies (*Labuz et al., 2007*; *Gaveriaux-Ruff et al., 2011*; *Weibel et al., 2013*; *Stein, 1993*; *Machelska et al., 1998*; *Brack et al., 2004*). No animals, samples, or outliers were excluded from analysis. Synthetic and chromatographic experiments were performed in biological replicates of N = 3. Behavioral and microdialysis experiments were performed in parallel and always with the corresponding controls. Each animal was used only once (behavioral tests or microdialysis). Data were analyzed by Sigma Plot software. Normality of data distribution was tested by Kolmogorov-Smirnov test. Changes over more than two time points induced by one treatment were evaluated using one-way repeated measurements (RM) analysis of variance (ANOVA) followed by Bonferroni test. Two-way RM ANOVA and Bonferroni test were used to compare two treatments or treatment versus control over time. Multiple comparisons at one time point were performed using one-way ANOVA followed by Bonferroni test for normally distributed data, and Kruskal-Wallis one-way ANOVA on ranks followed by Dunn test for non-normally distributed data. Dose-response relationships were tested by linear regression ANOVA. Paired or unpaired t-tests were used for two-sample comparisons of dependent or independent data, respectively. For radioligand binding data, nonlinear regression and calculation of the mean $IC_{50}$ was performed using Prism 5 (GraphPad). The difference between $IC_{50}$ values was evaluated using unpaired t-test. Differences were considered significant at values of $p < 0.05$.

## Acknowledgements

This study was supported by the Deutsche Forschungsgemeinschaft (DFG-MA 2437/2–1), the Bundesministerium für Bildung und Forschung (BMBF 0316177B), the European Society of Anaesthesiology (Research Grant 2010), the Helmholtz Virtual Institute 'Multifunctional Biomaterials for Medicine', and by the Focus Areas 'Nanoscale' and 'DynAge' of the Freie Universität Berlin and Charité. Dr. Leonie Lang contributed to drafting the application to the State Animal Ethics Committee.

## Additional information

### Funding

| Funder | Grant reference number | Author |
| --- | --- | --- |
| Deutsche Forschungsgemeinschaft | MA 2437/2-1 | Halina Machelska |
| Helmholtz-Gemeinschaft | | Rainer Haag |
| Bundesministerium für Bildung und Forschung | 0316177B | Christoph Stein |
| European Society of Anaesthesiology | Research Grant 2010 | Christoph Stein |

The funders had no role in study design, data collection and interpretation, or the decision to submit the work for publication.

### Author contributions

SG-R, Data curation, Formal analysis, Validation, Investigation, Visualization, Methodology, Writing—original draft, Project administration, Writing—review and editing; MAQ, Conceptualization, Formal analysis, Validation, Investigation, Visualization, Methodology, Writing—original draft, Project administration, Writing—review and editing; SG, Data curation, Formal analysis, Validation, Investigation, Visualization, Methodology, Writing—original draft, Writing—review and editing; KAW, Data curation, Formal analysis, Validation, Investigation, Visualization, Methodology, Writing—review and editing; XZ, MS, Validation, Investigation, Visualization, Methodology, Writing—original draft; VS,

Validation, Investigation, Visualization, Methodology, Writing—original draft, Writing—review and editing; DL, Formal analysis, Validation, Investigation, Visualization, Methodology, Writing—original draft; AR-G, Investigation, Methodology, Writing—review and editing; JJ, Validation, Investigation, Visualization, Methodology, Writing—review and editing; MKP, Conceptualization, Validation, Investigation, Visualization, Methodology, Writing—review and editing; HM, Data curation, Formal analysis, Supervision, Validation, Investigation, Visualization, Methodology, Writing—original draft, Project administration, Writing—review and editing; RH, Conceptualization, Data curation, Formal analysis, Supervision, Funding acquisition, Validation, Investigation, Visualization, Methodology, Writing—original draft, Project administration, Writing—review and editing; CS, Conceptualization, Resources, Data curation, Formal analysis, Supervision, Funding acquisition, Validation, Visualization, Writing—original draft, Project administration, Writing—review and editing

## Author ORCIDs

Viola Spahn, http://orcid.org/0000-0002-8086-8090
Dominika Labuz, http://orcid.org/0000-0003-2465-4730
Maria K Parr, http://orcid.org/0000-0001-7407-8300
Halina Machelska, http://orcid.org/0000-0001-6315-2958
Christoph Stein, http://orcid.org/0000-0001-5240-6836

## Ethics

Animal experimentation: The study was approved by the state ethics committee (Landesamt für Gesundheit und Soziales, Berlin, Germany; No. G0115/09) and was performed in strict accordance with the recommendations in the Guide for the Care and Use of Laboratory Animals of the National Institutes of Health.

## Additional files

**Supplementary files**
• Source data 1. Elemental analysis calculation-MQ.

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
