## [Decision Letter]

[Editors’ note: a previous version of this study was rejected after peer review, but the authors submitted for reconsideration. The first decision letter after peer review is shown below.]

Thank you for submitting your work entitled "Polyglycerol-opioid conjugates – a novel generation of painkillers designed to preclude side effects" for consideration by *eLife*. Your article has been reviewed by four peer reviewers, one of whom, Peggy Mason, is a member of our Board of Reviewing Editors and the evaluation has been overseen by Gary Westbrook as the Senior Editor. Our decision has been reached after consultation between the reviewers. Based on these discussions and the individual reviews below, we regret to inform you that your work will not be considered in its current form for publication in *eLife*.

First, it should be said that all reviewers appreciated the potential excitement around this new therapeutic approach. Thus, while we are rejecting this version, we are open to a new submission of a manuscript that addresses the key points.

1) First, the low pH of inflamed tissue is assuredly true but how low is unclear. Two references are given for this "fact." One (Stein and Machelska) contains no information on the pH of inflamed tissue. The other (Holzer) states " pH values can fall to 4.7 in fracture-related haematomas, to 5.4 in inflammation, to 5.7 in cardiac ischaemia, and to 6.2 during exhausting skeletal muscle contractions." Given this large range, the reviewers wanted to see a measurement of pH in the model used here (CFA).

2) Not unrelated to the first point, in vitro data on PG-M release as a function of pH and time is needed. This is particularly important because acidotic hydrolysis is not steeply dependent on pH and because of a highly plausible alternative possibility, namely that the PG-M release occurs through an esterase reaction. Whichever of these mechanisms causes release is obviously fine but the experiments need to be done to back up the conclusions.

There are additional points in the detailed reviews provided below which the authors may use to effect. However, the mechanism of action questions highlighted here are the central points that drove the decision to reject this manuscript in its current version.

*Reviewer #1:*

In general, I like this manuscript. The following are my major comments:

– The use of 12 mg/kg morphine for comparison is specious. As the authors demonstrate, this dose is sedating and constipating and causes respiratory depression to the extreme. It is above the dose that would be used clinically (in a non-tolerant pt). Saying that rats that have received 12 mg/kg morphine have no fecal boli is simply not that interesting or relevant.

– It is not accurate to say that the same amount of morphine resulted from the various opioid administration. The dose resulting from PG-M in injured paw may be equivalent to that yielded by M given systemically, but not this is not true of PG-M in the uninflamed tissue. The point that the amount of opioid in the inflamed paw is equivalent to that from systematic opioid is well taken without the frankly hyperbolic statement above.

– There is no figure legend for Figure 6.

– I don't understand the figure legend of Figure 7. Systemic doses are given for both iv PG-M and systemic morphine. Just give the actual doses used not the calculated doses.

*Reviewer #2:*

The manuscript by Gonzalez-Rodriguez and colleagues shows that the novel compound polyglycerol-morphine (PG-M) alleviates local inflammatory pain without producing sedation, a classically described morphine effect on the central nervous system, and with no constipatory action. The paper reports entirely new data as this is the first release on this new morphine polymer that gets cleaved to morphine in an acidic milieu only, a characteristic of inflammation sites. It is an important contribution in the fight against chronic pain that constitutes a major health problem worldwide. Indeed, opiates are the most potent analgesics but their use leads to important side effects that limit their utility. This work constitutes a key advance for the development of analgesics based on morphine self-degradable polymers that would not produce brain-mediated effects such as sedation or addiction. The data presented support the major conclusions and there are no concerns with ethics.

*Reviewer #3:*

This is an interesting manuscript with some fascinating data on the use of HPGs as a carrier for morphine coupled to the dendritic polymer by a biodegradable ester linkage. The synthesis of the material is sound and well supported. The biological results seem quite clear in that the material has an analgesic effect in the model examined. However, I have concerns about the mechanism proposed for the effect, as outlined below.

My issue is the mechanism of release of morphine and the supporting data to establish the acidic hydrolysis of the morphine substituents. It is known that acidic hydrolysis of esters is much less pH sensitive than basic hydrolysis, so a fairly acidic environment or long reaction times are generally recognized as necessary to provide significant bond rupture. Are these conditions satisfied in the inflammation model used? While the authors provide references reporting extreme acidic environments in some inflammatory conditions, the statements therein seem anecdotal and no supporting measurement data in these references is available. The best paper I found on this subject (A. Punnia-Moorthy, J. Oral Pathol 1987, 16: 36-44) describes direct pH measurements in the inflamed area and of the exudates in a model of inflammation induced by three agents, one bacterial as in the present case. The results show pH changes generally less than 0.5 pH units to ~pH 6.9-7.0 as the maximum drop with measurements up to 24 hr after injection. The authors do not show any data that would suggest this small change would be sufficient to release much morphine under their conditions. It would have helped to have in vitro data of release as a function of pH and time to look directly at this issue. I think what has been ignored here is the likelihood of the inflammatory exudate containing significant enzymatic esterase activity that could be responsible for ester hydrolysis and morphine release. Leukocyte esterases are routinely found in exudates and are used as evidence of the presence of inflammation clinically so it would be easy to test for this and to measure the release rates to compare with the acidic hydrolysis model.

*Reviewer #4:*

This is a well-written and very interesting manuscript with an innovative approach for the use of a polymer-drug conjugate. The in vivo experiments are really good. Only, a few more controls and physico-chemical characterization of PG-M conjugates would be needed in order to corroboration statement throughout the manuscript mainly referring to morphine pH-responsive release.

1) Characterization of drug loading and absence of encapsulated drug should be done in more quantitative manner by means for example of NMR- TOCSY would give you conjugation identity and DOSY could support absence of non-conjugated drug.

2) A systematic drug release kinetics study should be performed in vitro at different pHs, 7.4 and 4.7 as reported. Also including stability in plasma as local/sc vs. iv administrations are compared.

3) Also it would be nice to see PG-M accumulation in the inflamed areas by labeling PG with a NIR tag for example, this will give an idea of conjugate fate and will support clinical use if shows adequate PK/biodistribution data.

[Editors’ note: what now follows is the decision letter after the authors submitted for further consideration.]

Thank you for resubmitting your work entitled "Polyglycerol-opioid conjugate produces analgesia devoid of side effects" for further consideration at *eLife*. Your revised article has been favorably evaluated by Gary Westbrook (Senior Editor) and two reviewers, one of whom, Peggy Mason, is a member of our Board of Reviewing Editors.

The manuscript has been improved but there are some remaining issues that need to be addressed before acceptance, as outlined below. Both reviewers are skeptical of the importance of the acidic environment. And the cleaving of PG-MOR by a far more acidic environment than was measured is minimal. At this point, the whole acidic cleavage story appears to be a holdover from earlier drafts and constructions of the story but no longer appears warranted for inclusion given the new data. Or at the very least, the pH story needs to be toned down and background for the leukocyte esterase story expanded. More detailed comments are below and should be helpful in preparing a revision.

*Reviewer #1:*

I think this version is significantly improved and is much closer to acceptance. I am still unhappy with the discussion around the mechanism of morphine release from HPGs however. Measuring the local pH in vivo (6.8) and adding the release kinetics in vitro is helpful but I don't understand why the acidic pH chosen (5.5) is so much lower than the value measured in the inflamed rat paw. Based on the data presented I don't see any reason to expect pH to be involved in morphine release in vivo, at least not in this model. This comment is based on the very small and slow effect shown for more even acidic conditions than observed, shown in Figure 12, compared to esterase-driven release and to the very rapid (minutes) analgesic effects measured in vivo, shown in Figure 6. The authors' lose nothing by focussing on the esterase mechanism as it is universally recognized as a critical element of inflammation. As I said in my first review, it is known that acidic conditions are much less effective than basic ones in hydrolyzing esters so the short time scale of analgesia appearance observed in vivo should be an immediate sign that pH is *not* driving morphine release in the rat paw but it is quite consistent with what is known about the behavior of leukocytes in such circumstances. I would suggest then that:

a) A brief description of leukocyte esterase activity in inflammation should be provided in the Introduction or Discussion so the reader does not just abruptly come on the idea with no background, and;

b) A clearer discussion included comparing the results of Figure 12 and Figure 6 and what they mean mechanistically. I know the discussion is toned down with respect to pH effects compared to the original version. However, I don't think it is useful to the readership to not be clear that the enzymatic route is likely the important one and singularly promising for ongoing development of the exciting discovery resulting from this fine set of experiments. This is an important paper in my mind so mechanistic clarity is critical for future evolution of the approach.

*Reviewer #2:*

This revision adds two important findings. First, the authors have measured the actual pH in their injury model. Second they have shown that either low pH (far lower than measured and so of dubious relevance) or esterase (of far more relevance) will cleave the PG-MOR. This is a valuable contribution.

---

## [Author Response]

[Editors’ note: the author responses to the first round of peer review follow.]

*First, it should be said that all reviewers appreciated the potential excitement around this new therapeutic approach. Thus, while we are rejecting this version, we are open to a new submission of a manuscript that addresses the key points.*

*1) First, the low pH of inflamed tissue is assuredly true but how low is unclear. Two references are given for this "fact." One (Stein and Machelska) contains no information on the pH of inflamed tissue. The other (Holzer) states " pH values can fall to 4.7 in fracture-related haematomas, to 5.4 in inflammation, to 5.7 in cardiac ischaemia, and to 6.2 during exhausting skeletal muscle contractions." Given this large range, the reviewers wanted to see a measurement of pH in the model used here (CFA).*

We have now added in vivo pH measurements in paw tissue and blood, performed by a new co-author (A.R.G.). This information and a respective reference were added to the Results (subsections “PG-M produces analgesia selectively in inflamed tissue” and 2 In vivo concentrations of free morphine in paw tissue, circulating blood, and brain “), Discussion (paragraph five) and Materials and methods (“In vivo pH measurement”) sections. Additional references are now cited in Table 1.

*2) Not unrelated to the first point,* in vitro *data on PG-M release as a function of pH and time is needed. This is particularly important because acidotic hydrolysis is not steeply dependent on pH and because of a highly plausible alternative possibility, namely that the PG-M release occurs through an esterase reaction. Whichever of these mechanisms causes release is obviously fine but the experiments need to be done to back up the conclusions.*

We have now raised additional funds and added a substantial volume of new experiments to this effect. These were conducted and analyzed by three new co-authors (K.W., J.J., M.K.P.). We replaced Figure 3, added a new figure (Figure 12), removed previous Figure 9 (and related text. Several statements throughout the manuscript were altered to acknowledge the role of leukocyte esterase in addition to acidosis.) References were also updated/added.

*There are additional points in the detailed reviews provided below which the authors may use to effect. However, the mechanism of action questions highlighted here are the central points that drove the decision to reject this manuscript in its current version.*

*Reviewer #1:*

*In general, I like this manuscript. The following are my major comments:*

*– The use of 12 mg/kg morphine for comparison is specious. As the authors demonstrate, this dose is sedating and constipating and causes respiratory depression to the extreme. It is above the dose that would be used clinically (in a non-tolerant pt). Saying that rats that have received 12 mg/kg morphine have no fecal boli is simply not that interesting or relevant.*

We would like to retain this statement because these experiments impressively demonstrate that an equivalent dose of PG-M does not produce sedation, respiratory depression or constipation. This is one of the major findings of our study. In addition, a dose of 12 mg/kg morphine is not unusually high in rodents (e.g. Craft. Exp Clin Psychopharmacol. 2008;16:376-85; Nakamura et al., 2011; Bajic et al., 2015; Gallantine et al. Basic Clin Pharmacol Toxicol. 2008;103:419-27).

*– It is not accurate to say that the same amount of morphine resulted from the various opioid administration. The dose resulting from PG-M in injured paw may be equivalent to that yielded by M given systemically, but not this is not true of PG-M in the uninflamed tissue. The point that the amount of opioid in the inflamed paw is equivalent to that from systematic opioid is well taken without the frankly hyperbolic statement above.*

We agree with the reviewer. To avoid misunderstandings, we have changed the respective statements by replacing the terms “yield” with “contain” throughout the manuscript.

*– There is no figure legend for Figure 6.*

The figure legend is now included.

*– I don't understand the figure legend of Figure 7. Systemic doses are given for both iv PG-M and systemic morphine. Just give the actual doses used not the calculated doses.*

For clarification, the respective statements were amended in all figure legends.

*Reviewer #2:*

*The manuscript by Gonzalez-Rodriguez and colleagues shows that the novel compound polyglycerol-morphine (PG-M) alleviates local inflammatory pain without producing sedation, a classically described morphine effect on the central nervous system, and with no constipatory action. The paper reports entirely new data as this is the first release on this new morphine polymer that gets cleaved to morphine in an acidic milieu only, a characteristic of inflammation sites. It is an important contribution in the fight against chronic pain that constitutes a major health problem worldwide. Indeed, opiates are the most potent analgesics but their use leads to important side effects that limit their utility. This work constitutes a key advance for the development of analgesics based on morphine self-degradable polymers that would not produce brain-mediated effects such as sedation or addiction. The data presented support the major conclusions and there are no concerns with ethics.*

We thank the reviewer for their comments.

*Reviewer #3:*

*This is an interesting manuscript with some fascinating data on the use of HPGs as a carrier for morphine coupled to the dendritic polymer by a biodegradable ester linkage. The synthesis of the material is sound and well supported. The biological results seem quite clear in that the material has an analgesic effect in the model examined. However, I have concerns about the mechanism proposed for the effect, as outlined below.*

*My issue is the mechanism of release of morphine and the supporting data to establish the acidic hydrolysis of the morphine substituents. It is known that acidic hydrolysis of esters is much less pH sensitive than basic hydrolysis, so a fairly acidic environment or long reaction times are generally recognized as necessary to provide significant bond rupture. Are these conditions satisfied in the inflammation model used? While the authors provide references reporting extreme acidic environments in some inflammatory conditions, the statements therein seem anecdotal and no supporting measurement data in these references is available. The best paper I found on this subject (A. Punnia-Moorthy, J. Oral Pathol 1987, 16: 36-44) describes direct pH measurements in the inflamed area and of the exudates in a model of inflammation induced by three agents, one bacterial as in the present case. The results show pH changes generally less than 0.5 pH units to ~pH 6.9-7.0 as the maximum drop with measurements up to 24 hr after injection. The authors do not show any data that would suggest this small change would be sufficient to release much morphine under their conditions. It would have helped to have* in vitro *data of release as a function of pH and time to look directly at this issue. I think what has been ignored here is the likelihood of the inflammatory exudate containing significant enzymatic esterase activity that could be responsible for ester hydrolysis and morphine release. Leukocyte esterases are routinely found in exudates and are used as evidence of the presence of inflammation clinically so it would be easy to test for this and to measure the release rates to compare with the acidic hydrolysis model.*

We thank the reviewer for this valuable comment. We have now raised additional funds, performed new experiments (new Figure 12), and added a new table (Table 1) to this effect.

*Reviewer #4:*

This is a well-written and very interesting manuscript with an innovative approach for the use of a polymer-drug conjugate. The in vivo experiments are really good. Only, a few more controls and physico-chemical characterization of PG-M conjugates would be needed in order to corroboration statement throughout the manuscript mainly referring to morphine pH-responsive release.

1) Characterization of drug loading and absence of encapsulated drug should be done in more quantitative manner by means for example of NMR- TOCSY would give you conjugation identity and DOSY could support absence of non-conjugated drug.

We respectfully disagree that TOCSY or DOSY NMR would have provided more relevant data because these methods have the same level of precision as normal 1H NMR. Using three independent methods, we demonstrated that there was practically no free morphine present in PG-M. This was shown by UV-Vis spectroscopy, thin layer chromatography, gel permeation chromatography, and 1H NMR (Figure 2–Figure 5). The results of these experiments were mutually supportive of each other and clearly showed no evidence of free morphine (in the form of TLC spot, shift of UV absorption maximum and proton resonance signals corresponding to free drug). Hence, we did not see a compelling reason to perform DOSY or TOCSY-based experiments to repeat the same observations obtained from conventional characterization techniques.

*2) A systematic drug release kinetics study should be performed* in vitro *at different pHs, 7.4 and 4.7 as reported. Also including stability in plasma as local/sc vs. iv administrations are compared.*

We have now raised additional funds, performed such experiments, added a new figure (Figure 12). No free morphine was detectable in blood after i.v. PG-M, suggesting that PG-M is stable in the circulation (Figure 9). This now mentioned in the Discussion (paragraph five).

3) Also it would be nice to see PG-M accumulation in the inflamed areas by labeling PG with a NIR tag for example, this will give an idea of conjugate fate and will support clinical use if shows adequate PK/biodistribution data.

We agree with the reviewer that it would be nice to perform such additional experiments. Unfortunately, this would require stable radiolabeling, a significant number of additional animals, the approval of a new animal ethics protocol, and would exceed the scope of the present proof-of-concept study. We certainly hope that other groups (e.g. in the pharmaceutical industry) will be stimulated by our initial report and perform such studies in the framework of drug development.

[Editors' note: the author responses to the re-review follow.]

*The manuscript has been improved but there are some remaining issues that need to be addressed before acceptance, as outlined below. Both reviewers are skeptical of the importance of the acidic environment. And the cleaving of PG-MOR by a far more acidic environment than was measured is minimal. At this point, the whole acidic cleavage story appears to be a holdover from earlier drafts and constructions of the story but no longer appears warranted for inclusion given the new data. Or at the very least, the pH story needs to be toned down and background for the leukocyte esterase story expanded. More detailed comments are below and should be helpful in preparing a revision.*

We have now toned down the importance of acidic pH and emphasized the relevance of leukocyte esterase throughout the manuscript. Specifically, we made the following changes:

1) We removed (former) Table 1, Figure 5 and the respective text passages.

2) We emphasized the relevance of leukocyte esterase (and possibly other lytic enzymes) in the text (Introduction final paragraph; Discussion paragraph five) and discussed the difference between in vitro and in vivo release of morphine from the conjugate.

3) The passage on characterization of the PG-M conjugate was removed from the Materials and methods section because this was already described under Results.